# Trajectory-aware Shifted State Space Models for Online Video Super-Resolution

**Qiang Zhu**[1], **Xiandong Meng**[1*], **Yuxuan Jiang**[2], **Fan Zhang**[2], **David Bull**[2]
**Shuyuan Zhu**[3*], **Bing Zeng**[3], **Ronggang Wang**[1,4*]
[1]Pengcheng Laboratory,  [3]University of Electronic Science and Technology of China
[2]University of Bristol,  [4]Shenzhen Graduate School, Peking University
{zhuqiang,mengxd}@pcl.ac.cn
eezsy@uestc.edu.cn, rgwang@pkusz.edu.cn

## Abstract

Online video super-resolution (VSR) is an important technique for many real-world video processing applications, which aims to restore the current high-resolution video frame based on temporally previous frames. Most of the existing online VSR methods solely employ one neighboring previous frame to achieve temporal alignment, which limits long-range temporal modeling of videos. Recently, state space models (SSMs) have been proposed with linear computational complexity and a global receptive field, which significantly improve computational efficiency and performance. In this context, this paper presents a novel online VSR method based on **T**rajectory-aware **S**hifted **SSMs** (**TS-Mamba**), leveraging both long-term trajectory modeling and low-complexity Mamba to achieve efficient spatio-temporal information aggregation. Specifically, TS-Mamba first constructs the trajectories within a video to select the most similar tokens from the previous frames. Then, a Trajectory-aware Shifted Mamba Aggregation (TSMA) module consisting of proposed shifted SSMs blocks is employed to aggregate the selected tokens. The shifted SSMs blocks are designed based on Hilbert scannings and corresponding shift operations to compensate for scanning losses and strengthen the spatial continuity of Mamba. Additionally, we propose a trajectory-aware loss function to supervise the trajectory generation, ensuring the accuracy of token selection when training our model. Extensive experiments on three widely used VSR test datasets demonstrate that compared with six online VSR benchmark models, our TS-Mamba achieves state-of-the-art performance in most cases and over 22.7% complexity reduction (in MACs). The source code for TS-Mamba is available at https://github.com/QZ1-boy/TS-Mamba.

## 1 Introduction

Among various video super-resolution (VSR) application scenarios, online VSR has recently attracted significant interest due to the growing popularity of live video conferencing and live broadcasting applications (Fuoli et al., 2023; Xiao et al., 2023). In online VSR, the current high-resolution (HR) video frame is typically restored using only its low-resolution (LR) counterpart and previous frames. This is constrained by the requirements for low latency and low computational complexity inherent to these online real-time applications.

In a VSR model, temporal alignment or aggregation is a core module employed to compensate for the information from neighboring frames before generating the current HR frame. Advanced temporal alignment or aggregation modules have been recently developed, which are based on deformable convolution networks (DCN) (Wang et al., 2019; Tian et al., 2020), flow-guided deformable alignment models (Chan et al., 2022a; Zhu et al., 2024c), non-local attention mechanisms (Li et al., 2020; Yi et al., 2019), Vision Transformer based spatio-temporal information aggregation (Liu et al., 2022a; Tang et al., 2023) or Diffusion models (Wang et al., 2025; Liu et al., 2025; Zhuang et al., 2025).

---

*Corresponding Authors.

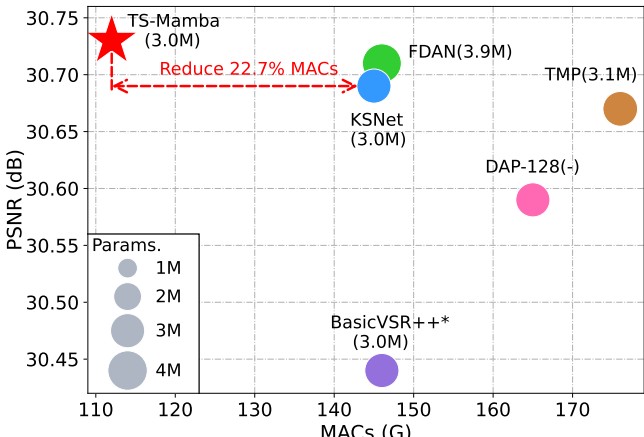

Figure 1: Comparison of existing online VSR methods with our TS-Mamba in terms of PSNR and MACs on the REDS4 dataset. Our TS-Mamba outperforms these SOTA methods and significantly reduces complexity in terms of MACs.

Although they offer superior VSR performance, these methods are typically associated with high complexity and, therefore, are not ideal for online VSR.

To mitigate these limitations, recent online VSR methods have adopted more efficient temporal alignment modules, such as lightweight optical flow networks (Sajjadi et al., 2018; Xiao et al., 2023), deformable attention mechanisms (Fuoli et al., 2023; Yang et al., 2023), and temporal motion propagation modules (Zhang et al., 2024b). For example, CKBG (Xiao et al., 2023) utilized a lightweight optical flow network to estimate motion between frames and perform motion compensation. DAP (Fuoli et al., 2023) designed a deformable attention pyramid module to dynamically focus on the most salient locations between frames and progressive refine the offsets to improve temporal alignment performance. FDAN (Yang et al., 2023) proposed a flow-guided deformable attention propagation module that introduces the optical flow to guide the offset generation to efficiently exploit the temporal information between frames. Despite their efficiency, these methods predominantly use short-term temporal information based on convolutional neural networks (CNN) — typically from a single previous frame, which restricts their ability to further enhance reconstruction quality. While incorporating long-term temporal alignment can improve performance, it often introduces significant computational overhead, resulting in challenges for real-time or resource-constrained applications. Therefore, it is valuable to develop the efficient long-range models for online VSR applications.

Recently, low-complexity state space models (SSMs) (Gu et al., 2021; Gu & Dao, 2023) have been proposed with linear computational complexity and with relatively large receptive fields, which can potentially improve performance with limited complexity. Inspired by this, we propose a Trajectory-aware Shifted Mamba for online VSR, denoted as **TS-Mamba**, leveraging long-term trajectory modeling and low-complexity Mamba for achieving the token-level spatio-temporal aggregation. In TS-Mamba, trajectories within a video are first constructed for selecting the most similar tokens from the previous frames. A trajectory-aware shifted Mamba aggregation (TSMA) module is then employed that consists of shifted SSMs blocks to aggregate the selected tokens. The shifted SSMs blocks are designed with specific procedures based on Hilbert scannings and four shift operations to compensate for scanning losses and strengthen the spatial continuity of Mamba. Moreover, we propose a trajectory-aware loss function to supervise the trajectory generation, optimizing the accuracy of token selection when training our model. The proposed TS-Mamba model enables efficient long-term video modeling with significantly reduced computational complexity. The primary contributions are summarized as follows:

- TS-Mamba is the **first SSMs-based online VSR model**, which aggregates long-term spatio-temporal information from previous frames at the token level for restoring current HR frame. This is different from existing online VSR methods which typically use CNN-based temporal alignment to exploit temporal information from a single previous frame.
- This is also the **first time to introduce video trajectories** into Mamba to select the most similar tokens from previous frames and construct the new trajectory-aware shifted Mamba model for efficient token-level spatio-temporal information aggregation.

- The **novel shifted SSMs blocks** are designed based on four different shift operations and Hilbert scannings to effectively compensate for the intra-window and inter-window losses of Hilbert scannings and strengthen the local spatial continuity of Mamba.

The proposed method has been benchmarked on three widely used test datasets and shows superior VSR performance with more than 22.7% computational complexity reduction in terms of MACs over five state-of-the-art (SOTA) online VSR methods (as shown in Figure 1).

## 2 RELATED WORK

### 2.1 VIDEO SUPER-RESOLUTION

Video super-resolution (VSR) is a fundamental low-level vision task that aims to restore an HR video from its LR counterpart. Existing VSR methods are typically learning-based, utilizing various deep neural networks (Teed & Deng, 2020; Zhu et al., 2019; Arnab et al., 2021; Ho et al., 2022). For example, optical flow-based methods (Chan et al., 2021; Liu et al., 2022b) explore the temporal motion between frames to align them; deformable convolution networks (DCN)-based methods (Tian et al., 2020; Wang et al., 2019; Dong et al., 2023) learn the motion offsets between frames for feature alignment. Moreover, flow-guided deformable-based methods (Chan et al., 2022a; Zhu et al., 2024c) combine optical flow and DCN to achieve better feature alignment. Non-local attention-based methods (Li et al., 2020; Yi et al., 2019) aggregate global information for feature aggregation. Vision Transformer-based methods (Liu et al., 2022a; Tang et al., 2023; Lin et al., 2022) aggregate long-term spatio-temporal information in video to restore SR frames. Diffusion-based methods (Wang et al., 2025; Liu et al., 2025; Zhuang et al., 2025) build the long-range modeling using designed diffusion models for information aggregation. However, these methods are still associated with high complexity and are therefore not best suited for real-time online VSR applications. Recently, some Mamba-based works (Xiao & Wang, 2025; Tran et al., 2025) were proposed that use low-complexity Mamba with global receptive filed to improve VSR performance. Although Mamba-based methods achieve some reduction in model complexity, their high-overhead repeated scannings hinders efficient implementation of real-time online VSR.

### 2.2 ONLINE VIDEO SUPER-RESOLUTION

Due to the specific requirements of online applications, online VSR methods are expected to be lightweight and have low latency. Therefore, most existing online VSR methods have been proposed (Sajjadi et al., 2018; Cao et al., 2021; Fuoli et al., 2023; Xiao et al., 2023; Jiang et al., 2025a) with efficient feature alignment modules. For example, EGVSR (Cao et al., 2021) and CKBG (Xiao et al., 2023) utilized lightweight optical flow networks to estimate motion between frames and perform motion compensation. KSNet (Jin et al., 2023) proposed a kernel-split manner to reparameterize convolutional kernels on the high-value channel, enabling representation of dynamic information and reducing complexity along the channel dimension. TMP (Zhang et al., 2024b) employs an efficient temporal motion propagation method that leverages motion field continuity to achieve fast feature alignment. DAP (Fuoli et al., 2023) designed a deformable attention pyramid module to dynamically focus on the most salient locations between frames and progressive refine the offsets to achieve temporal alignment improvement. FDAN (Yang et al., 2023) proposed a flow-guided deformable attention propagation module that introduces the optical flow to guide the offset generation to efficiently exploit the temporal information between frames. It is noted that, however, these online VSR methods are only based on one previous frame in feature alignment due to the complexity limitation, which hinders further improvement of VSR performance. Different from existing deformable-based methods (Fuoli et al., 2023; Yang et al., 2023) that based on short-term spatio-temporal aggregation, in this work, our TS-Mamba introduces of trajectories and designs shifted SSMs blocks, enabling it improve the ability for long-range spatio-temporal information aggregation.

### 2.3 STATE SPACE MODELS

State space models (Gu et al., 2021; Gu & Dao, 2023), e.g., Mamba, have been widely employed in vision tasks (Liu et al., 2024; Zhu et al., 2024a) due to their linear computational complexity and ability to model global dependencies. Recently, some Mamba-based methods are proposed for

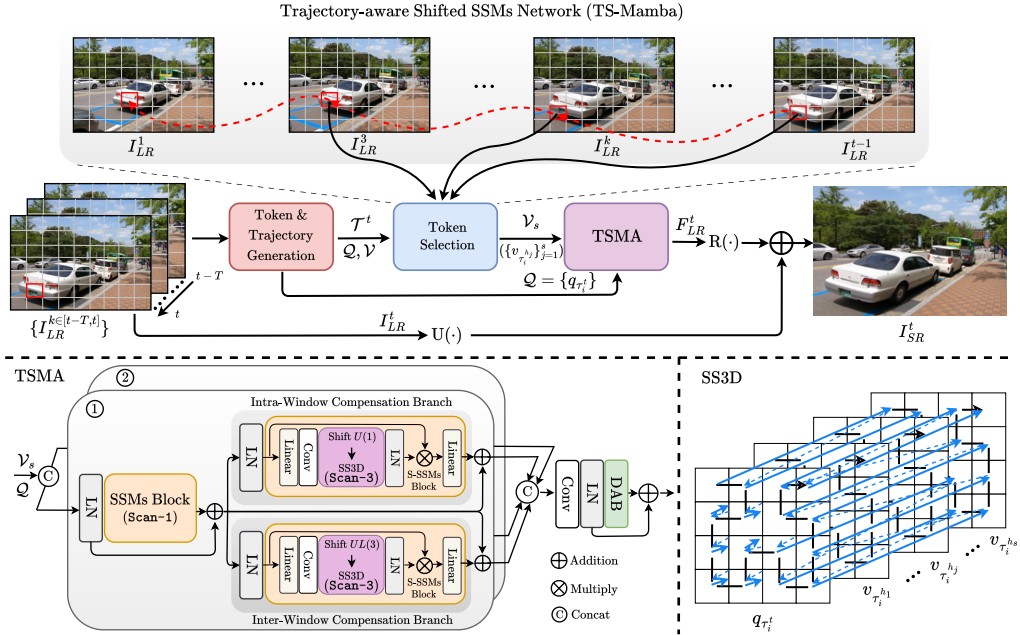

Figure 2: The architecture of the TS-Mamba network. Trajectories of videos are first generated and the similar tokens from previous frames are selected along trajectories. Then, the selected tokens alongside the current frame token are fed into the trajectory-aware shifted Mamba aggregation (TSMA) module to achieve the long-term spatio-temporal information aggregation.

image/video super-resolution. For example, MambaIR (Guo et al., 2024), and MambaIRv2 (Guo et al., 2025) used Mamba to achieve the global receptive field. TAMambaIR (Peng et al., 2025) introduced a texture-aware state space model to focus on textures regions for improving SR quality. VSRM (Tran et al., 2025) proposed the spatial-to-temporal Mamba and the temporal-to-spatial Mamba to ability of spatio-temporal aggregation. MamEVSR (Xiao & Wang, 2025) proposed a interleaved Mamba and a cross modality Mamba to interleave tokens and further leverage spatio-temporal information to capture finer details. Typically, Mamba converts 2D images into 1D tokens through scanning (Qiao et al., 2024; Shi et al., 2025), resulting in spatial continuity loss inherent to images. To enhance the ability of Mamba, some advanced scanning techniques have emerged to address this issue, such as bidirectional scanning (Hu et al., 2024; Shi et al., 2025), cross scanning (Liu et al., 2024), continuous 2D scanning (Yang et al., 2024a), and local scanning (Huang et al., 2024). To the best of our knowledge, the use of Mamba has not yet been investigated for the online video super-resolution task. Unlike existing Mamba-based works (Xiao & Wang, 2025; Tran et al., 2025) that neglect the local spatial continuity of Mamba, we introduce sophisticated shift operations for Hilbert scannings to enhance the ability of Mamba to maintain local spatial continuity, improving the online VSR performance with high efficiency.

## 3    METHODOLOGY

In online video super-resolution, when reconstructing the $t^{\text{th}}$ frame in a low-resolution video, we denote the current LR frame as $I_{LR}^t$ and temporally previous LR frames as $\{I_{LR}^k, k \in [t-T, t-1]\}$. The proposed trajectory-aware shifted state space models, TS-Mamba, are illustrated in Figure 2. Here, all these LR video frames $\{I_{LR}^k, k \in [t-T, t]\}$ are first fed into the token and trajectory generation $\text{G}(\cdot)$ module to extract the current frame token $\mathcal{Q}$ and the tokens of previous LR frames $\mathcal{V}$:

$$\mathcal{Q} = \text{G}\left(I_{LR}^t\right) = \left\{q_i^t\right\}, i \in [1, N], \tag{1}$$

$$\mathcal{V} = \text{G}\left(\{I_{LR}^k\}\right) = \left\{v_i^k\right\}, i \in [1, N], k \in [t-T, t-1], \tag{2}$$

where $\text{G}(\cdot)$ consists of a convolution layer and $N_1$ residual blocks to generate features and tokens from video frames, $N$ is the token number, and $T$ is the temporal window size. Based on the generated tokens $\{q_i^t\}$, the trajectories $\mathcal{T}^t$ of $I_{LR}^t$ frame can be formulated as a set of trajectories,

$$\mathcal{T}^t = \left\{\tau_i^k = \left(x_i^k, y_i^k\right)\right\}, i \in [1, N], k \in [t-T, t], \tag{3}$$

where $x_i^k \in [1, H]$, $y_i^k \in [1, W]$, and $H$ and $W$ represent the height and width of the feature (for LR frame), respectively. Each trajectory $\tau_i^k$ contains a sequence of coordinates $\{(x_i^k, y_i^k), i \in [1, N]\}$, and the end point of trajectory $\tau_i^t$ is associated with the coordinate $(x_i^t, y_i^t)$ of token $q_i^t$.

We then select $s$ the most similar tokens $\mathcal{V}_s$ along the trajectories and feed them into the proposed trajectory-aware shifted Mamba aggregation (TSMA) module alongside token $\mathcal{Q}$ to achieve spatio-temporal information aggregation:

$$F_{LR}^t = \text{TSMA}(\mathcal{Q}, \mathcal{V}_s). \tag{4}$$

Finally, the aggregated feature $F_{LR}^t$ and current LR frame $I_{LR}^t$ are fed into the reconstruction network $\text{R}(\cdot)$ and the upsampling $\text{U}(\cdot)$ network, respectively, to produce the super-resolved frame $I_{SR}^t$:

$$I_{SR}^t = \text{R}(F_{LR}^t) + \text{U}(I_{LR}^t), \tag{5}$$

in which $\text{R}(\cdot)$ consists of two convolution layers, $N_2$ residual blocks, and a pixelshuffle layer. $\text{U}(\cdot)$ here represents the bicubic upsampling operation.

## 3.1 TOKEN SELECTION

In order to select the most similar tokens along trajectories, we first reformulate tokens $\mathcal{Q}$, $\mathcal{V}$ associated with trajectories $\mathcal{T}^t$. Based on the formulation of the trajectories in Equation 3, tokens $\mathcal{Q}$, and $\mathcal{V}$ can be formulated as:

$$\begin{aligned} \mathcal{Q} &= \{q_{\tau_i^t}\}, \ i \in [1, N], \\ \mathcal{V} &= \{v_{\tau_i^k}\}, \ i \in [1, N], \ k \in [t - T, t - 1]. \end{aligned} \tag{6}$$

We compute cosine similarity between the token $\mathcal{Q}$, and tokens $\mathcal{V}$ to select $s$ the most similar tokens along trajectories. The indices of the selected tokens and the selected tokens can be formulated as:

$$\begin{aligned} \{h_j\}_{j=1}^s &= \underset{k}{\text{Top-k}} \langle \frac{q_{\tau_i^t}}{\| q_{\tau_i^t} \|_2^2}, \frac{v_{\tau_i^k}}{\| v_{\tau_i^k} \|_2^2} \rangle, \ h_j \in [1, T-1], \\ \mathcal{V}_s &= \{v_{\tau_i^{h_j}}\}_{j=1}^s, \ i \in [1, N]. \end{aligned} \tag{7}$$

Thus, the process of the TS-Mamba network is described as:

$$\begin{aligned} I_{SR}^t &= \text{TS-Mamba}(\mathcal{Q}, \mathcal{V}, \mathcal{T}^t) \\ &= \text{R}(\underset{\tau_i^k \in \mathcal{T}^t}{\text{TSMA}}(q_{\tau_i^t}, \{v_{\tau_i^{h_j}}\}_{j=1}^s)) + \text{U}(I_{LR}^t). \end{aligned} \tag{8}$$

## 3.2 TRAJECTORY-AWARE SHIFTED MAMBA AGGREGATION

Mamba networks are typically used to convert 2D images into 1D tokens via scanning, resulting in spatial continuity losses inherent to the images. Existing works (Zhang et al., 2024a; Xiao & Wang, 2025) do not analyze the degree of discontinuous regions but instead repeatedly use multiple scannings, making these methods hard to maintain the spatial continuity of the image and instead lead to greater complexity.

To address this issue, in this work, we first analyzed the spatial discontinuity in Hilbert scannings and then proposed a trajectory-aware shifted Mamba aggregation (TSMA) module that combines a standard SSMs block and the proposed shifted SSMs (S-SSMs) blocks in the "Scan-Shift-Scan" manner to compensate for the intra-window and inter-window losses of Hilbert scannings. As illustrated in Figure 2, in the TSMA module, token $\mathcal{Q}$ and selected tokens $\mathcal{V}_s$ are first concatenated along the channel dimension and fed into two paths in a specific "Scan-Shift-Scan" manner, i.e., ① or ②, each of which consists of a standard SSMs block and two parallel S-SSMs blocks to compensate for the losses according to the scanning of the standard SSMs block. The output of each path is concatenated and then aggregated by a convolution layer and a deformable attention block (DAB) (Xia et al., 2022) to obtain the output feature. Each SSMs/S-SSMs block and DAB is preceded by layer normalization (LN) and is followed by a residual connection. In each SSMs/S-SSMs block, the trajectory-aware tokens are scanned based on spatial Hilbert selective scannings along the temporal dimension (SS3D) to capture long-term spatio-temporal characteristics.

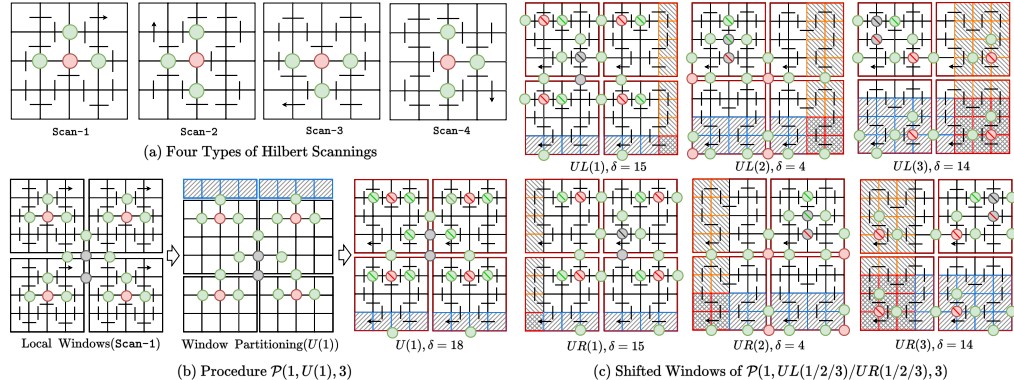

Figure 3: Illustration of Hilbert scannings and shifted windows generated by seven procedures. (a) Four types of Hilbert scannings. (b) The procedure $\mathcal{P}(1, U(1), 3)$, and elimination value $\delta$. (c) Shifted windows and elimination values $\delta$ for procedures $\mathcal{P}(1, UL(1/2/3)/UR(1/2/3), 3)$, respectively.

### 3.3 DISCONTINUITY FOR HILBERT SCANNING

To evaluate the spatial discontinuity of Hilbert scannings in local windows, we define the discontinuity degree $D_d$ as follows. If the four adjacent areas are successively scanned, the region consisting of these four scanned areas is considered as continuous ($D_d = 0$); otherwise, the discontinuity degree $D_d$ equals the number of areas that are not successively scanned. For a region consisting of four adjacent areas, the range of discontinuity degree is $D_d \in \{0, 1, 2, 3\}$. This is illustrated in Figure 3 (a), where four typical Hilbert scannings, i.e., $\mathcal{G}_{Scan} = \{\texttt{Scan-1}, \texttt{Scan-2}, \texttt{Scan-3}, \texttt{Scan-4}\}$, are shown on a 4×4 grid. Here the region with $D_d = 1$ is marked by a green circle and the region with $D_d = 2$ is marked by a red circle.

Moreover, we extend the general case to that based on the 8×8 grid to further discuss the discontinuity degrees. An 8×8 region is partitioned into four 4×4 local regions and we illustrate the discontinuity degree $D_d$ within and between local windows under $\texttt{Scan-1}$ in Figure 3 (b). It can be observed that both intra-window discontinuity and inter-window discontinuity exist. In particular, due to the nature of Hilbert scanning, the central region between windows is widely spaced (the inter-window discontinuity), resulting in inter-level gaps. Here, the discontinuity degree $D_d$ equals 3 - we mark this region with a gray circle in Figure 3 (b).

### 3.4 SHIFTED SSMS BLOCK

To eliminate the discontinuity of Hilbert scannings, we propose the "Scan-Shift-Scan" manner that combines window shifting with specific Hilbert scannings to strengthen the continuity of SSMs. The shifting can be defined based on the shift direction and shift position, e.g., Up 1 position ($U(1)$), Up Left 1 position ($UL(1)$) and Down Right 2 position ($DR(2)$). Our "Scan-Shift-Scan" manner is designed based on the four Hilbert scannings (shown in Figure 3 (a)) and these window shifting processes. As shown in Figure 3 (b), we illustrate the procedure of $\texttt{Scan-1} \rightarrow U(1) \rightarrow \texttt{Scan-3}$ as an example. The local windows are first partitioned by $U(1)$ shift operation and then cyclic fed as the shifted windows. It can be inferred that the second scanning ($\texttt{Scan-3}$) on the shifted window can eliminate the discontinuity of first scanning ($\texttt{Scan-1}$). To simplify the description of procedure, we define the procedure as:

$$\mathcal{P}(l, \mathcal{S}f(p), j) = \mathcal{S}c_1(l) \rightarrow \mathcal{S}f(p) \rightarrow \mathcal{S}c_2(j), \tag{9}$$

where the first and the second scannings are denoted as $\mathcal{S}c_1(l), \mathcal{S}c_2(j) \in \mathcal{G}_{Scan}$, $l, j \in \{1, 2, 3, 4\}$. Shift operations are denoted as $\mathcal{S}_f(p) \in \{U(p), UL(p), UR(p), D(p), DL(p), DR(p), p \in \{1, 2, 3\}\}$. Therefore, procedure $\texttt{Scan-1} \rightarrow U(1) \rightarrow \texttt{Scan-3}$ can be formulated as $\mathcal{P}(1, U(1), 3)$.

To evaluate the discontinuity elimination, we set three symbols and define an elimination value $\delta$ to mark and calculate the elimination. Specifically, we use "\", "\", and "\" on the circle for representation that eliminates 1, 2, and 3 discontinuity degrees, respectively, in Figure 3 (b)-(c). The elimination value $\delta$ is calculated by summing the eliminated discontinuity degrees that consist of intra-window discontinuity elimination and inter-window discontinuity elimination, i.e.,

$\delta = \delta_{\text{intra}} + \delta_{\text{inter}}$. We have investigated possible procedures and illustrated the representative shifted windows generated by six shift operations, i.e., $UL(1)$, $UL(2)$, $UL(3)$, and $UR(1)$, $UR(2)$, $UR(3)$, under the first scanning (Scan-1) and second scanning (Scan-3) in Figure 3 (c). It can be inferred from Figure 3 (b)-(c) that procedure $\mathcal{P}(1, U(1), 3)$ achieves the best elimination ($\delta$=18), and the best intra-window discontinuity elimination ($\delta_{\text{intra}}$=18) but doesn't eliminate inter-window discontinuity ($\delta_{\text{inter}}$=0). We infer that the other three procedures also achieve the best elimination: $\mathcal{P}(2, L(1), 4)$, $\mathcal{P}(3, D(1), 1)$, $\mathcal{P}(4, R(1), 2)$. Moreover, procedures $\mathcal{P}(1, UL(3), 3)$, and $\mathcal{P}(1, UR(3), 3)$ have the best inter-window discontinuity elimination ($\delta_{\text{inter}}$=6) but worse than procedure $\mathcal{P}(1, U(1), 3)$ for intra-window discontinuity elimination ($\delta_{\text{intra}}$=8). The more procedures and details are provided in the supplementary.

Based on the formula of procedure, we can calculate the elimination value $\delta$ of the a procedure $\mathcal{P}(l, \mathcal{S}f(p), j)$. We summary all the procedures with our supplementary and we can infer its range value of elimination value $\delta \in [4, 18]$, the range value of intra-window discontinuity elimination $\delta_{\text{intra}} = [0, 18]$ and the range value of inter-window discontinuity elimination $\delta_{\text{inter}} = [0, 6]$. Therefore, we elaborately find out the combinations of shift operations and Hilbert scannings to construct two S-SSMs blocks in parallel branches, i.e., intra-window compensation branch (IntraWCB) and inter-window compensation branch (InterWCB), to optimally eliminate corresponding discontinuities. As illustrated in Figure 2, we set two procedures for the parallel SSMs blocks to construct our TSMA module: ①: $\mathcal{P}(1, U(1), 3) + \mathcal{P}(1, UL(3), 3)$; ②: $\mathcal{P}(2, L(1), 4) + \mathcal{P}(2, LU(3), 4)$ to achieve sufficient elimination of discontinuity.

### 3.5 SELECTIVE SCANNING ALONG TEMPORAL DIMENSION

To achieve temporal token aggregation, we implement spatial Hilbert-based selective scanning along the temporal dimension, i.e., SS3D. As shown in Figure 2, we showcase the SS3D processing with Scan-1. The current token $\{q_{\tau_i^t}\}$ and selected tokens $\{v_{\tau_i^{h_j}}\}_{j=1}^s$ are scanned to convert spatio-temporal neighboring pixels into a 1D token sequence. Each token sequence undergoes selective scanning based on the local windows. This process interweaves selected tokens with current tokens, enabling information to interact across spatial and temporal dimensions to capture long-term spatio-temporal characteristics. By scanning spatio-temporally adjacent pixels, SS3D preserves local spatial information and progressively captures global temporal patterns.

### 3.6 LOSS FUNCTION

We adopt Charbonnier loss (Lai et al., 2018) as the spatial loss function to supervise the SR frame generation:

$$\mathcal{L}_{spa} = \sqrt{\|I_{HR}^t - I_{SR}^t\|^2 + \epsilon^2}, \tag{10}$$

in which $I_{HR}^t$ is the HR frame and the $\epsilon$ is set to $1 \times 10^{-4}$.

To supervise the trajectory generation for ensuring the accuracy of token selection, we first employ the formulation of trajectories of LR video in Equation 3 to generate trajectories of HR video:

$$\mathcal{T}_{HR}^t = \left\{ \tau_{i(HR)}^k = \left( x_i^k, y_i^k \right) \right\}, i \in [1, M], k \in [t - T, t]. \tag{11}$$

Based on this, we propose our trajectory-aware loss function:

$$\mathcal{L}_{trj} = \left\| \mathcal{T}^t - ((\mathcal{T}_{HR}^t) \downarrow_{\hat{s}})/\hat{s} \right\|, \tag{12}$$

where $\downarrow_{\hat{s}}$ is the downsampling with scale factor $\hat{s}$ that subsamples every $\hat{s}$ coordinate to LR size.

Overall, the total loss is:

$$\mathcal{L}_{total} = \mathcal{L}_{spa} + \lambda \mathcal{L}_{trj}, \tag{13}$$

in which the hyperparameter $\lambda$ is set to 0.1.

## 4 EXPERIMENTS

### 4.1 EXPERIMENTAL SETTINGS

Following previous online VSR research (Jin et al., 2023; Zhang et al., 2024b), we use REDS (Nah et al., 2019), and Vimeo-90K (Xue et al., 2019) as training datasets. REDS4 is used for evaluating the

Table 1: Comparison with state-of-the-art online VSR methods. The runtime, FPS, parameters, and PSNR(dB)/SSIM are reported on three benchmarks with BI and BD degradations.

| Category | Methods | Supp. Frame | R-T ≈ | Run.↓ (ms) | FPS↑ (1/s) | MACs↓ (G) | Params.↓ (M) | BI degradation REDS4(RGB)↑ (PSNR/SSIM) | BI degradation Vid4(Y)↑ (PSNR/SSIM) | BD degradation Vimeo-90K-T(Y)↑ (PSNR/SSIM) | BD degradation Vid4(Y)↑ (PSNR/SSIM) |
|---|---|---|---|---|---|---|---|---|---|---|---|
| Bidirectional | BasicVSR | P+F | ✗ | 63 | 15.9 | 397 | 6.3 | 31.42/0.8909 | 27.24/0.8251 | 37.53/0.9498 | 27.96/0.8553 |
| | IconVSR | P+F | ✗ | 70 | 14.3 | 452 | 8.7 | 31.67/0.8948 | 27.39/0.8279 | 37.84/0.9524 | 28.04/0.8570 |
| | BasicVSR++ | P+F | ✗ | 77 | 13.0 | 418 | 7.3 | 32.39/0.9069 | 27.79/0.8400 | 38.21/0.9550 | 29.04/0.8753 |
| | SSL-bi | P+F | ✗ | 24 | 41.7 | 92 | 1.0 | 31.06/0.8933 | 27.15/0.8208 | 37.06/0.9458 | 27.56/0.8431 |
| | DFVSR | P+F | ✗ | - | - | - | 7.1 | 32.76/0.9081 | 27.92/0.8427 | 38.51/0.9571 | 29.56/0.8983 |
| | MIA-VSR | P+F | ✗ | 318 | 3.1 | 3220 | 16.5 | 32.78/0.9220 | 28.20/0.8507 | - | - |
| | IART | P+F | ✗ | 180 | 5.6 | 5020 | 13.4 | 32.90/0.9138 | 28.26/0.8517 | 38.62/0.9579 | 29.68/0.8884 |
| | VSRM | P+F | ✗ | 223 | 4.5 | 2174 | 17.1 | 33.11/0.9162 | 28.44/0.8552 | - | - |
| Online | Bicubic | N | ✓ | - | - | - | - | 26.14/0.7292 | 23.78/0.6347 | 31.30/0.8687 | 21.80/0.5246 |
| | RRN | P | ✓ | 34 | 29.4 | 193 | 3.4 | 28.82/0.8234 | 25.85/0.7660 | 36.69/0.9432 | 27.69/**0.8488** |
| | BasicVSR++* | P | ✓ | 40 | 25.0 | 146 | **3.0** | 30.44/0.8686 | 27.06/0.8173 | 37.11/0.9464 | 27.49/0.8426 |
| | DAP-128 | P | ✓ | 38 | 26.3 | 165 | - | 30.59/0.8703 | - | 37.29/0.9476 | - |
| | FDAN | P | ✓ | 34 | 29.4 | 146 | 3.9 | 30.71/0.8723 | 27.14/0.8206† | **37.36**/0.9483† | **27.76**/0.8471 |
| | KSNet | P | ✓ | 31 | 32.3 | 145 | **3.0** | 30.69/0.8724 | 27.14/0.8208 | 37.34/**0.9490** | 27.63/0.8444† |
| | TMP | P | ✓ | **25** | **40.1** | 176 | 3.1 | 30.67/0.8710 | 27.10/0.8167 | 37.33/0.9481 | 27.61/0.8428 |
| | VSRM* | P | ✓ | 31 | 32.7 | 136 | 3.1 | 30.64/0.8701 | 27.10/0.8163 | 37.28/0.9477 | 27.57/0.8423 |
| | **TS-Mamba** | P | ✓ | 29 | 33.5 | **112** | **3.0** | **30.73/0.8727** | **27.17/0.8209** | **37.36**/0.9482 | 27.70/0.8473 |

models trained on the REDS dataset, while Vimeo-90K-T and Vid4 (Liu & Sun, 2013) are utilized for benchmarking the models trained on the Vimeo-90K dataset. Two degradations, BI (bicubic) and blur degradation (BD), are used to perform downsampling and the downsampling factor is set to $\hat{s} = 4$. For BI downsampling, the HR frame is downsampled by a bicubic filter. For BD downsampling, the HR frame is first blurred by a Gaussian filter with standard deviation $\sigma = 1.6$, and then the blurred frame is subsampled for every $\hat{s}$ pixels to generate the LR frame. PSNR and SSIM are adopted as performance evaluation metrics. Runtime (Run.), FPS (frames per second), MACs, and parameters (Params.) are computed on an LR frame of size $180{\times}320$ to evaluate model complexity and speed.

In the experiments, the numbers of residual blocks $N_1$ and $N_2$ are set to 2 and 13, respectively. The token size is $4{\times}4$ and the window size is $8{\times}8$. The selected token number $s$ is set as 3. Random flips, rotations, and temporal inversion operations are performed for data augmentation. Adam optimizer (Kingma, 2014), and Cosine Annealing scheme (Loshchilov & Hutter, 2016) are used during network training. The HR patch size is $256{\times}256$ and the batch size is 8. The total number of iterations is 600K. The proposed method is implemented on the PyTorch platform with two NVIDIA GeForce RTX 3090 GPUs. Following (Liu et al., 2022a), a lightweight optical flow network (Kong et al., 2021) is adopted to update trajectories. The temporal window size $T$ is set as 15 based on (Zhang et al., 2024b) when training on REDS (Nah et al., 2019). For the Vimeo-90K (Xue et al., 2019) dataset, the original sequence is temporally flipped to obtain a 14-frame sequence.

We compare our approach with five SOTA online VSR methods, including RRN (Isobe et al., 2020), DAP-128 (Fuoli et al., 2023), FDAN (Yang et al., 2023), KSNet (Jin et al., 2023), and TMP (Zhang et al., 2024b), and eight bidirectional propagation VSR methods, BasicVSR (Chan et al., 2021), IconVSR (Chan et al., 2021), BasicVSR++ (Chan et al., 2022a), SSL (Xia et al., 2023), DFVSR (Dong et al., 2023), MIA-VSR (Zhou et al., 2024b), IART (Xu et al., 2024) and VSRM (Tran et al., 2025)). Additionally, we implemented another methods, i.e.,"BasicVSR++*" and "VSRM*", by removing the backward propagation branch of VSR models, i.e., BasicVSR++ and VSRM, and reducing its model size for online VSR application. We use "P","F" and "N" to represent those with the previous support (supp.) frames, future support frames and no support frames.

## 4.2 OVERALL PERFORMANCE

As shown in Table 1, the quantitative results demonstrate the superior performance of the proposed method over other online VSR models in terms of PSNR and SSIM. We also supplement the results of FDAN and KSNet models on Vid4 and Vimeo-90K-T datasets based on their released pre-trained models and source codes for a comprehensive comparison. These results are reported in Table 1 with

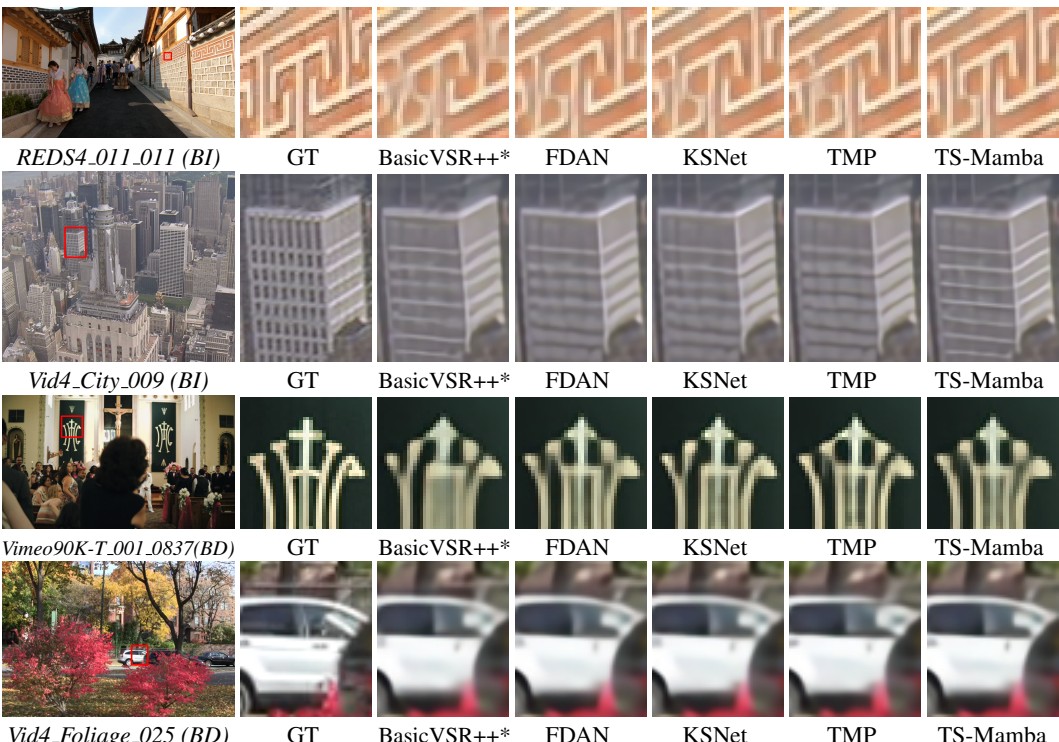

Figure 4: Visual results on BI degradation (REDS4, Vid4) and BD degradation (Vimeo-90K-T, Vid4).

Table 2: Results of the ablation study on REDS4 dataset.

| Models | PSNR(dB)↑ / SSIM↑ | Params.(M)↓ | Run.(ms)↓ | MACs(G)↓ |
|---|---|---|---|---|
| (v1.1) *w/o* Trajectory | 30.45 / 0.8678 | 1.7 | 20 | 84 |
| (v1.2) *w/o* $\mathcal{L}_{trj}$ | 30.70 / 0.8721 | 3.0 | 29 | 112 |
| (v1.3) *w/o* IntraWCB | 30.58 / 0.8702 | 2.8 | 25 | 97 |
| (v1.4) *w/o* InterWCB | 30.61 / 0.8706 | 2.8 | 25 | 97 |
| (v1.5) *w/o* IntraWCB+InterWCB | 30.52 / 0.8689 | 2.4 | 21 | 85 |
| (v1.6) *w/o* $U(1)/D(1)$ | 30.65 / 0.8710 | 3.0 | 27 | 112 |
| (v1.7) *w/o* $UL(3)/DL(3)$ | 30.67 / 0.8714 | 3.0 | 27 | 112 |
| (v1.8) *w/o* (v1.6) + (v1.7) | 30.61 / 0.8702 | 3.0 | 25 | 111 |
| **TS-Mamba (ours)** | 30.73 / 0.8727 | 3.0 | 29 | 112 |

"†". Figure 4 presents qualitative comparisons, from which we can observe that our method shows better visual quality than other online VSR methods for both BI and BD degradations.

Following (Fuoli et al., 2023; Zhang et al., 2024b), VSR methods that can process 720p (1280×720) videos in at least 24 in terms of FPS are recognized as real-time (R-T.) methods (Fuoli et al., 2023), and we have labelled all the tested methods in Table 1 according to their runtime. It is noted that our TS-Mamba model achieves the second fastest inference speed among all online VSR methods. TMP is the one with the fastest runtime as it was implemented with the CUDA accelerator (high MACs but low runtime) while TS-Mamba is not. Moreover, TS-Mamba also offers a significant reduction in terms of MACs (about 36.3%) and a marginal reduction in parameter numbers compared to TMP, as shown in Figure 1.

## 4.3 ABLATION STUDY

To further verify the effectiveness of our contributions, we have conducted ablation studies on the REDS4 dataset with BI degradation.

We first confirmed the contribution of two trajectory-aware designs, i.e., trajectory generation and trajectory-aware loss, by creating the following variants. (v1.1) *w/o* Trajectory - G(·) and Token Selection module were removed from TS-Mamba; (v1.2) *w/o* $\mathcal{L}_{trj}$ - the trajectory-aware loss function

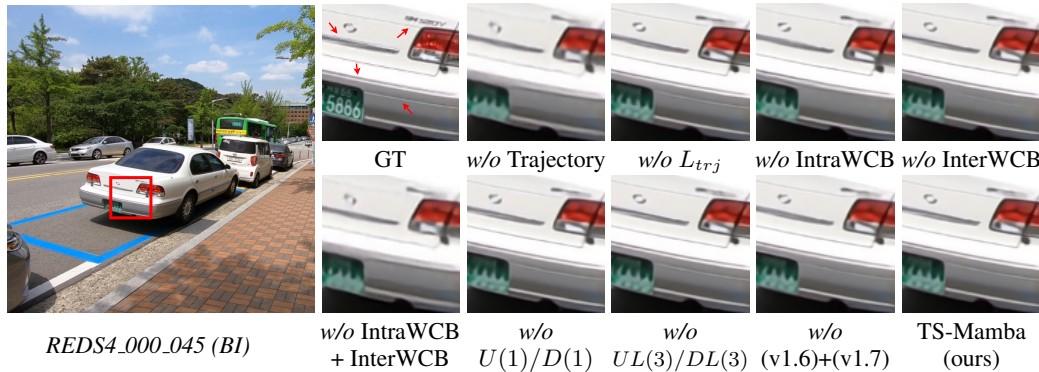

Figure 5: Visual results of the ablation study on REDS4 dataset.

| $s$ | PSNR(dB)↑/SSIM↑ | Params.↓ (M) | Run.↓ (ms) | MACs↓ (G) |
|---|---|---|---|---|
| 1 | 30.64/0.8712 | 2.8 | 25 | 96 |
| 2 | 30.68/0.8720 | 2.9 | 27 | 104 |
| 3 | 30.73/0.8727 | 3.0 | 29 | 112 |
| 4 | 30.74/0.8727 | 3.1 | 31 | 120 |

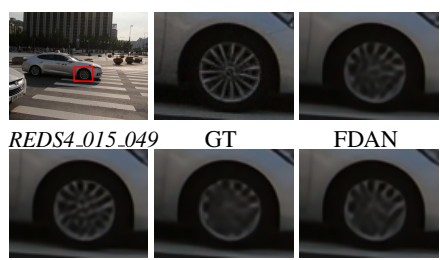

Table 3: Ablation study of selected token number $s$.

Figure 6: A failure case on REDS4 dataset.

was removed when training the TS-Mamba model. We further verified our proposed TSMA module in terms of compensation branches and shift operations, by obtaining (v1.3) *w/o* IntraWCB and (v1.4) *w/o* InterWCB - IntraWCB and InterWCB were removed from the TSMA module, respectively; (v1.5) *w/o* IntraWCB + InterWCB - both IntraWCB and InterWCB were removed from the TSMA module. We also tested the adopted shift operations in compensation branches, and implemented (v1.6) *w/o* $U(1)/L(1)$ - the $U(1)/L(1)$ shift operations were removed in IntraWCB, (v1.7) *w/o* $UL(3)/LU(3)$ - the $UL(3)/LU(3)$ shift operations were removed in InterWCB and (v1.8) *w/o* (v1.6)+(v1.7) - all the shift operations were removed in TSMA module. As shown in Table 2, the performance of all these variants is evidently lower than that of the full TS-Mamba, which fully confirms the effectiveness of each key component in our design. Moreover, we further provide the visual results of these variants on REDS4 dataset in Figure 5. It is found that the visual results demonstrates the contributions of our designs, particularly in realistic textures and fine details of the car.

To confirm the value of the token number $s$ in our TS-Mamba, we tested different $s$ values with our TS-Mamba, and presented the results in Table 3. It is noted that as $s$ increases, the VSR performance improves, but with higher model complexity. When $s = 4$, it is difficult to obviously improve VSR performance. To trade off between complexity and performance, we set $s = 3$ in this work.

## 5 LIMITATIONS

We investigate our results and find out the failure cases. A failure case when highly dynamic rotation occurs is visualized in Figure 6. The generated trajectories and compensated manner of our TS-Mamba are inaccurate enough when dynamic rotation occurs in car tire and rotation information cannot be reconstructed, thus limiting the performance of our method. Due to the high difficulty of modeling rotation, other online VSR methods also fail to obtain complete rotation information.

## 6 CONCLUSION

In this paper, we proposed a **T**rajectory-aware **S**hifted **SSMs** (**TS-Mamba**) for online VSR, leveraging long-term trajectory modeling and low-complexity Mamba to achieve efficient spatio-temporal information aggregation. In TS-Mamba, trajectories in a video are first constructed to select the most similar tokens from the previous frames. A trajectory-aware shifted Mamba aggregation module is then employed, which consists of shifted SSMs blocks to aggregate the selected tokens. The shifted SSMs blocks are designed based on Hilbert scannings and shift operations to compensate for the scanning losses and strengthen the spatial continuity of Mamba. Extensive experiments on three widely used VSR benchmarks have demonstrated the effectiveness and efficiency of our method.

ACKNOWLEDGMENTS

This work was supported by the Key Research and Development Program of Peng Cheng Laboratory under Grant PCL2024A02, the China Scholarship Council, the University of Bristol, and the UKRI MyWorld Strength in Places Programme (SIPF00006/1).

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

## A    MORE ANALYSIS FOR TSMA MODULE

In this section, we provide a full analysis of the evidence of our procedures in our TSMA module. Four Hilbert scannings (as shown in Figure 7) and ten shift operations ($U(1)$, $D(1)$, $L(1)$, $R(1)$, $UL(1/2/3)$, $UR(1/2/3)$) are adopted to determine the suitable procedures for the best elimination performance.

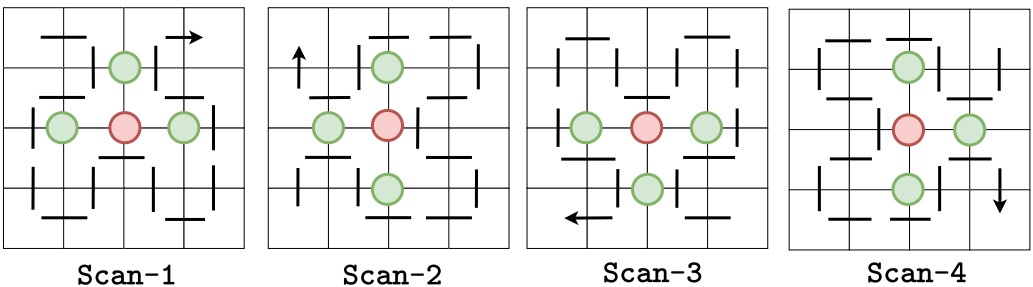

Figure 7: Four types of Hilbert scannings.

We first provide the procedures of previous methods (Zhang et al., 2024a; Xiao & Wang, 2025), i.e., multiple scannings with no shift operations, to prove the necessity of shift operations. Four procedures of previous methods under `Scan-1` and their corresponding elimination values $\delta$ are illustrated in Figure 8:
`Scan-1`→ `Scan-1`; `Scan-1`→ `Scan-2`;
`Scan-1`→ `Scan-3`; `Scan-1`→ `Scan-4`.
It can be found from Figure 8 that the discontinuity of `Scan-1` cannot be eliminated when using `Scan-1` as the second scanning. A few of the intra-window discontinuity can be eliminated and the inter-window discontinuity cannot be eliminated when using `Scan-2`/`Scan-3`/`Scan-4` as the second scanning. These results imply that using multiple scannings on local windows makes it hard to eliminate the discontinuity of Hilbert scans.

Based on this problem, we introduce the shift operations under Hilbert scannings to enhance the discontinuity elimination. To find suitable shift operations for specific scanning to achieve the best elimination performance. We attempt some procedures under first scanning is `Scan-1` with four different shift operations ($U(1)$, $D(1)$, $L(1)$, $R(1)$) to determine the second scanning. These procedures are illustrated in Figure 9. We can find from Figure 9 that:
`Scan-1`→$U(1)$→`Scan-3` has the best elimination performance ($\delta$=18).
`Scan-1`→$U(1)$→`Scan-1`/`Scan-2`/`Scan-4` have the same elimination value ($\delta$=14).
`Scan-1`→$D(1)$→`Scan-1`/`Scan-2`/`Scan-3`/`Scan-4` have the same elimination value ($\delta$=14).
`Scan-1`→$L(1)$→`Scan-1`/`Scan-2`/`Scan-3`/`Scan-4` have the same elimination value ($\delta$=14).
`Scan-1`→$R(1)$→`Scan-1`/`Scan-2`/`Scan-3`/`Scan-4` have the same elimination value ($\delta$=14).
From these results, it can be inferred that the procedure (1): `Scan-1`→$U(1)$→`Scan-3` is the most suitable procedure.

Based on the observation, we extend the procedure (1) to other first scannings to construct the other three procedures:
(2):`Scan-2`→$L(1)$→`Scan-4`;
(3):`Scan-3`→$D(1)$→`Scan-1`;
(4):`Scan-4`→$R(1)$→`Scan-2`.
These four procedures and their elimination values $\delta$ are illustrated in Figure 10. It can be inferred that these four procedures can obtain the best elimination performance. Besides, we also found that these procedures have obvious symmetry. Furthermore, the $UL(1/2/3)$ and $UR(1/2/3)$ shift operations also have the same symmetry. The procedures of the $UL(1/2/3)$ and $UR(1/2/3)$ shift operations are illustrated in the Figure. 3 (c) in our paper (also shown in Figure 11 in this file). Therefore, in our paper, we have determined to use the procedures (1) and (2) to construct our shifted SSMs blocks in the TSMA module. It is noted that although the second Hilbert scanning brings new discontinuous areas, these areas have already been overcome in the first scanning because the discontinuous areas do not overlap when using the two Hilbert scannings.

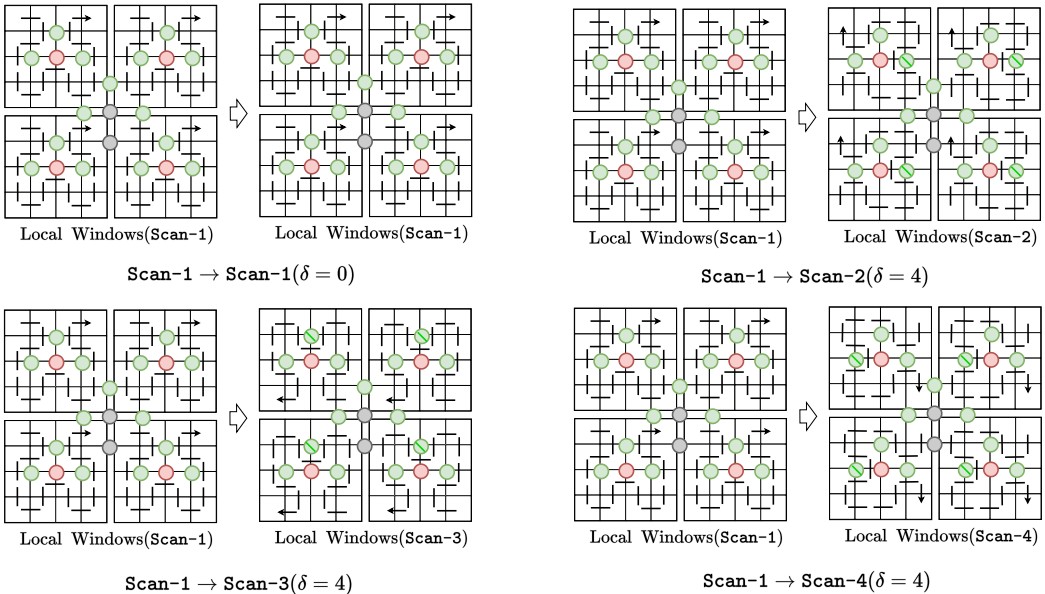

Figure 8: Four procedures of previous methods and the corresponding elimination values.

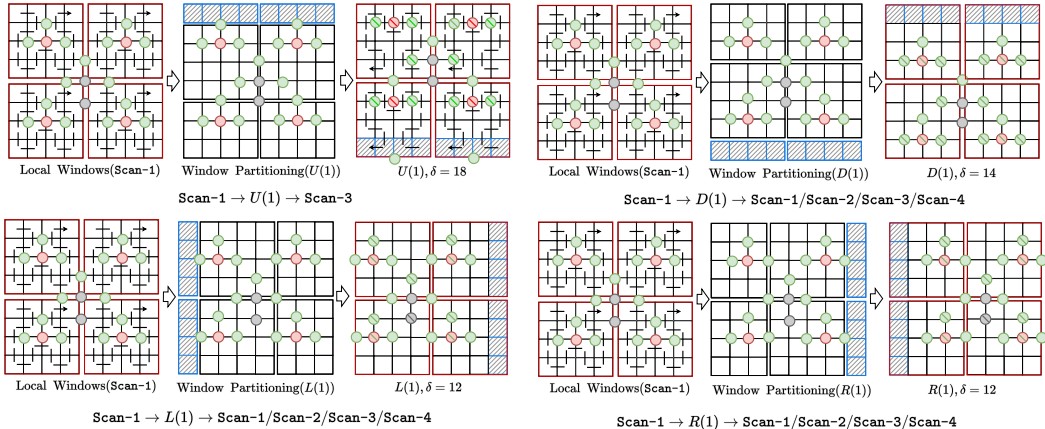

Figure 9: Determining suitable procedures under first scanning `Scan-1` and four shift operations.

# B ADDITIONAL EXPERIMENTS

In this section, we conduct the additional experiments of ablation study and compared methods for a comprehensive comparison.

## B.1 ADDITIONAL ABLATION STUDIES

*(1) Ablation Study of Different Deformable-based Modules for Our TS-Mamba.* We implement these alignments, i.e., deformable convolution network (DCN), flow-guided deformable alignment (FGDA), deformable attention (DA), and flow-guided deformable attention (FDA) to replace the TSMA module in our TS-Mamba for comparison. Their corresponding results are reported in Table 4. It can be see that our TSMA module achieves the best PSNR/SSIM performance and the lowest complexity, which demonstrates the effectiveness of our TSMA module.

*(2) Ablation Study of Temporal Window Size.* We set different temporal window size $T$ from 3 to 23 to determine the optimal size of temporal window and the corresponding results are provided in Table 5. Besides, previous works also use the $T$=15 as the fixed window size, in our experiment,

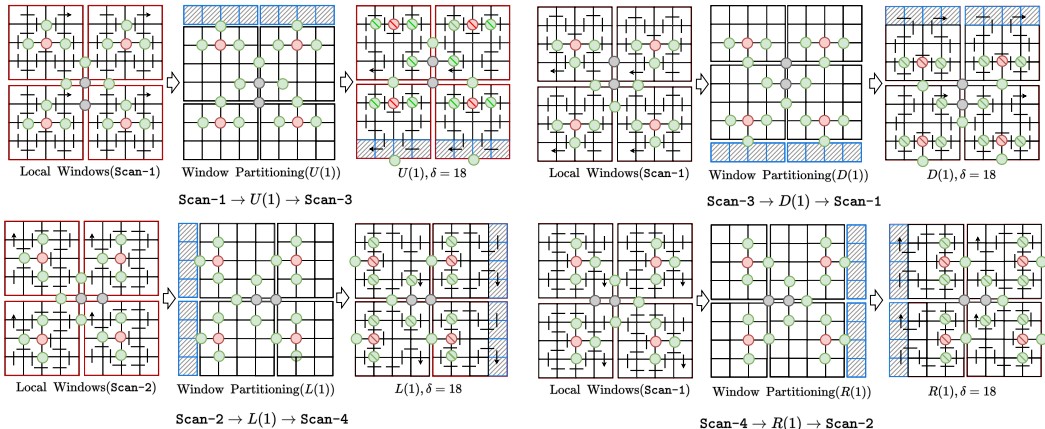

Figure 10: Four procedures and corresponding elimination $\delta$.

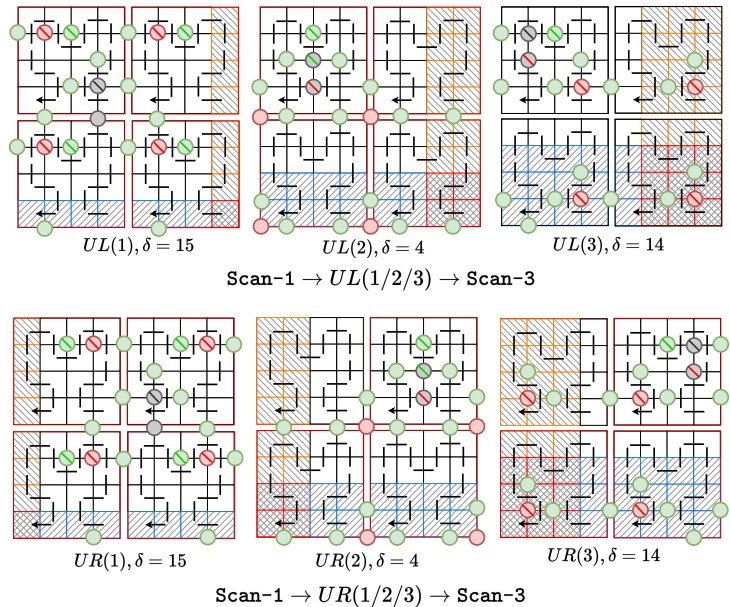

Figure 11: Six procedures and elimination value $\delta$ for Scan-1 $\rightarrow UL(1/2/3)/UR(1/2/3) \rightarrow$ Scan-3.

for fair comparison, we also set our temporal window size $T$ as 15. From Table 5 see that when $T$ is increase, the PSNR is increase, which indicates that longer temporal information can help optimize the designed models and achieve the better VSR performance.

## B.2 ADDITIONAL COMPARISONS

*(1) Comparisons of Mamba-based Methods.* We first discuss the designs of MambaIR (Shi et al., 2025), MambaIRv2 (Guo et al., 2025), TAMambaIR (Peng et al., 2025), and VSRM (Tran et al., 2025) with our TS-Mamba. MambaIR proposes a residue state space block with local enhancement and channel redundancy reduction based on four directions scannings to boost the restoration ability of Mamba. MambaIRv2 designs an attentive state-space model with a semantic-guided neighboring mechanism to encourage strong interaction between pixels under a single direction scanning, which effectively eliminating the high complexity and redundancy of multi-directional scans in existing methods. TAMambaIR introduces a texture-aware state space model that modulates the transition

Table 4: Comparison with different alignments with our TS-Mamba.

| Models | PSNR(dB)↑/SSIM↑ | Params.(M)↓ | Run.(ms)↓ | MACs(G)↓ |
|---|---|---|---|---|
| DCN | 30.59/0.8696 | 2.8 | 30 | 132 |
| FGDA | 30.64/0.8701 | 3.1 | 42 | 148 |
| DA | 30.67/0.8706 | 3.0 | 33 | 145 |
| FDA | 30.69/0.8714 | 3.0 | 31 | 142 |
| TSMA (ours) | 30.73/0.8727 | 3.0 | 29 | 112 |

Table 5: Comparison of different temporal window sizes with our TS-Mamba.

| Models | PSNR(dB)↑/SSIM↑ | Params.(M)↓ | Run.(ms)↓ | MACs(G)↓ |
|---|---|---|---|---|
| $T = 3$ | 30.57/0.8698 | 2.4 | 23 | 96 |
| $T = 7$ | 30.65/0.8713 | 2.7 | 25 | 103 |
| $T = 11$ | 30.70/0.8722 | 2.8 | 27 | 107 |
| $T = 15$ (ours) | 30.73/0.8727 | 3.0 | 29 | 112 |
| $T = 19$ | 30.73/0.8730 | 3.3 | 31 | 118 |
| $T = 23$ | 30.74/0.8731 | 3.7 | 33 | 123 |

matrix of Mamba with multi-directional perception blocks and focus on regions with complex textures to enhance texture awareness. In the VSR task, VSRM introduces a spatial-to-temporal Mamba and a temporal-to-spatial Mamba blocks based bidirectional scannings to extract long-range spatio-temporal features and enhance receptive fields efficiently. These Mamba-based methods usually use the multiply scannings or extra interaction module to enhance the ability of Mamba but they neglect the local spatial continuity of Mamba, which cause the limited SR performance. Our TS-Mamba analyzes the local spatial discontinuity and definites the degree of discontinuity and combines the Hilbert scannings and shift operations to obtain strength the ability of Mamba. Notes that Hilbert scannings have the better spatial continuity than bidirectional scannings and cross scannings, and introduce of shift operations is a simple yet efficient way that strength the local spatial continuity of Mamba and without increasing the complexity, which is effectively help our TS-Mamba achieves the high efficiency spatio-temporal information aggregation.

Table 6: Comparison of Mamba-based methods with our TS-Mamba on REDS4 dataset.

| Category | Methods | PSNR(dB)↑/SSIM↑ | Params.(M)↓ | MACs(G)↓ |
|---|---|---|---|---|
| Image SR | MambaIR | 32.25/0.9019 | 20.42 | 779.7 |
| | MambaIRv2 | 32.48/0.9054 | 23.10 | 1567.2 |
| | TAMambaIR | - | - | - |
| | MambaIR-light | 26.89/0.8195 | 0.92 | 84.6 |
| | MambaIRv2-light | 27.36/0.8389 | 0.79 | 75.6 |
| Video SR | VSRM | 33.11/0.9162 | 17.1 | 2174 |
| | VSRM* | 30.64/0.8701 | 3.1 | 136 |
| | TS-Mamba (ours) | 30.73/0.8727 | 3.0 | 112 |

To evaluate these methods with our TS-Mamba, we conduct experiments of MambaIR, MambaIRv2, and VSRM on REDS4 dataset with BI degradation and implement a transformed method, i.e.,"VSRM*", by removing the backward propagation branch of VSRM and reducing its model size for online VSR application. Their results are provided in Table 6. It is found that Mamba-based VSR methods has better SR performance than Mamba-based ISR methods, which due to the temporal information lose impacts the restoration quality. Besides, VSRM has the best performance than other methods but its high complexity leads to suitable for online real-time applications. Compared with two Mamba-based online methods, i.e., VRSM* and TS-Mamba, our TS-Mamba achieves the high PSNR and low complexity, which due to the long-range spatio-temporal information aggregation

Table 7: Comparison of Mamba-based image SR methods.

| Methods | Set5 PSNR/SSIM | BSDS100 PSNR/SSIM | Manga109 PSNR/SSIM | Params. (M)↓ | MACs (G)↓ |
|---------|----------------|-------------------|--------------------|--------------| ----------|
| MambaIR | 38.57/0.9627 | 32.58/0.9048 | 40.28/0.9806 | 20.42 | 221.0 |
| TAMambaIR | 38.58/0.9627 | 32.58/0.9048 | 40.35/0.9810 | 16.07 | 180.0 |
| MambaIRv2 | 38.65/0.9631 | 32.62/0.9053 | 40.42/0.9810 | 22.90 | 445.8 |

Table 8: Comparison different RealVSR methods with our TS-Mamba on RealVSR dataset.

| Methods | ILNIQE↓/NRQM↑ | Params.(M)↓ | Run.(ms)↓ |
|---------|---------------|-------------|-----------|
| RealVSR | 34.39/3.795 | 2.7 | 772 |
| BasicRealVSR | 30.37/6.582 | 6.3 | 73 |
| RealViformer | 28.61/6.588 | 5.3 | 49 |
| VSRM | 30.29/6.613 | 17.1 | 223 |
| VSRM* | 33.29/4.368 | 3.1 | 31 |
| TS-Mamba (ours) | 32.54/5.161 | 3.0 | 29 |

and shifted Mamba compensation, while VRSM* only use two previous frames to aggregate temporal information, limiting the exploration of long-range spatio-temporal information. Due to the TAMambaIR method without releasing its source code, we cannot compared it with other methods. To compare it with other methods as much as possible, we collect the image SR results of three image SR methods from their papers as a reference and are provided in Table 7. It can be found that TAMambaIR reduces the complexity compared with MambaIR and MambaIRv2, but it did not exceed MambaIRv2 in SR performance.

*(2) Comparisons of Real-world VSR Methods.* We conduct a latest Mamba-based VSR model, i.e., VSRM, its variant VSRM* and our TS-Mamba model on RealVSR dataset for evaluation on real-world scenarios. For further making a comprehensive comparison, we adopted three representative real-world VSR methods, i.e., RealVSR (Yang et al., 2021), BasicRealVSR (Chan et al., 2022b), RealViformer (Zhang & Yao, 2024) on RealVSR dataset, the corresponding experimental results of compared methods and our method are provided in Table 8. It is found that since no noise or unknown complex degradations were introduced during the training process, the general VSR results are not as effective as the RealVSR methods. Compared to the RealVSR, our TS-Mamba achieves better VSR results while maintaining lower complexity, making it a potential replacement for RealVSR.

*(3) Comparisons of Recent VSR Methods.* Recently, some methods use advanced structures such as CNN (Meng et al., 2019; 2020; Zhu et al., 2022; Qiu et al., 2023; Zhu et al., 2024e; Yang et al., 2024b; Zhu et al., 2024d;b; Xiao et al., 2021; Zhu et al., 2025; Jiang et al., 2025b), Transformer (Liu et al., 2022a; Zhu et al., 2023; Zhou et al., 2024b; Ren et al., 2025; Li et al., 2025) and diffusion models (Zhou et al., 2024a; Yang et al., 2024c), to achieve the promising VSR performance. Here, we discuss some representative methods and compared them with our TS-Mamba.

Some diffusion-based VSR methods use diffusion for long-range information modeling to achieve the superior generation performance. For example, LiftVSR (Wang et al., 2025) introduces a hybrid temporal modeling mechanism that decomposes temporal learning into dynamic temporal attention (DTA) and attention memory cache (AMC). DTA for fine-grained temporal modeling within short frame segment and AMC for long-term temporal modeling across segments. UltraVSR (Liu et al., 2025) proposes a lightweight recurrent temporal shift module that by partially shifting feature components along the temporal dimension to enable effective propagation, fusion, and alignment across frames without explicit temporal layers. Additionally, it introduces a temporally asynchronous inference strategy to capture long-range temporal dependencies under limited memory constraints. FlashVSR (Zhuang et al., 2025) proposes a diffusion-based one-step real-time VSR that consists a train-friendly three stage distillation pipeline, a locality constrained sparse attention that adopted a KV-cache to maintain the spatio-temporal consistency and preserve the high fidelity of videos.

Table 9: Comparison of long-range modeling diffusion-based methods with TS-Mamba.

| Methods | PSNR↑ | SSIM↑ | Param.(M)↓ | Run.(ms)↓ |
|---|---|---|---|---|
| LiftVSR | 24.34 | - | - | - |
| UltraVSR | 24.50 | 0.6962 | 10.5 | 89 |
| FlashVSR | 24.11 | 0.6511 | 1780.14 | 15500 |
| TS-Mamba (ours) | 30.73 | 0.8727 | 3.1 | 29 |

Table 10: Comparison with different general VSR methods with our TS-Mamba on REDS4 dataset.

| Category | Methods | PSNR(dB)↑/SSIM↑ | Params.(M)↓ | Run.(ms)↓ | MACs(G)↓ |
|---|---|---|---|---|---|
| CNN-based | DFVSR | 32.76/0.9081 | 7.1 | - | - |
| Transformer-based | S2SVR | 31.96/0.8988 | 13.4 | 194 | 3462 |
| | MIA-VSR | 32.78/0.9220 | 16.5 | 318 | 3220 |
| | IA-RT | 32.90/0.9138 | 13.4 | 180 | 5020 |
| Mamba-based | VSRM | 33.11/0.9162 | 3.0 | 223 | 2174 |
| | VSRM* | 30.64/0.8701 | 3.0 | 31 | 136 |
| | TS-Mamba (ours) | 30.73/0.8727 | 3.0 | 29 | 112 |

We provide the results of these methods and our TS-Mamba on REDS4 dataset in Table 9. It is found that our TS-Mamba model achieve the best PSNR and SSIM performance with a large margin while a significant efficiency. These recent works use the all long-range information in videos for current frame reconstruction which brings high complexity while our TS-Mamba selects the most similar token based on the trajectories to aggregate the long-range spatio-temporal information, which avoiding to all information processing so as to reduce model complexity for online VSR.

Besides of diffusion-based methods, some CNN-based and Transformer-based methods also proposed in the VSR task. For example, A CNN-based method, i.e., DFVSR (Dong et al., 2023), proposed a directional frequency representation and a directional frequency-enhanced alignment to represent the property of frequency of detail and direction information, and use double enhancements of task-related information to generate the high-quality feature. S2SVR (Lin et al., 2022) proposes an sequence-to-sequence model with an unsupervised optical flow estimator to maximize its potential in capturing long-range dependencies among frames. With reliable optical flow, the accurate correspondence is established among multiple frames for improving the restoration performance. We added their results and two latest Transformer-based VSR methods, i.e., MIA-VSR (Zhou et al., 2024b) and IA-RT (Xu et al., 2024) on REDS4 dataset with BI degradation in Table 10. It is found that although S2SVR, DFVSR, MIA-VSR and IA-RT achieves the better VSR performance than out TS-Mamba model, but they have the high complexity and low inference speed, which cannot apply into the real-time online VSR processing.

These two VSR methods uses the consecutive one or two frames to achieve the temporal alignment or aggregation, while our TS-Mamba model utilizes the long-range spatio-temporal information based on trajectories in video from all the previous frames for spatio-temporal aggregation. Duo to the trajectory-aware aggregation, our TS-Mamba can more easy to obtain the better VSR performance.

*(4) Comparisons of Real-time SR Methods.* Recently, some real-time SR methods were proposed. For comprehensive comparison with real-time SR methods, we compare these methods in quantitative or empirical comparison. EGVSR (Cao et al., 2021) designed a lightweight CNN network structure based on spatio-temporal adversarial learning and efficient upsampling method to reduce the computation and guarantee the high visual quality. RTSR (Jiang et al., 2025a) utilizes a dual teacher knowledge distillation network for optimization of compressed content at various quantization levels to achieve the low-complexity SR. Different from these two real-time methods, our TS-Mamba introduces the low-complexity and global receptive field Mamba with long-range temporal trajectories to achieve the long-range spatio-temporal aggregation while these two models only use CNN network with local receptive field and neglect the long-rang temporal information, which limits they further improve SR

Table 11: Comparison of real-time methods with TS-Mamba on Vid4 dataset.

| Methods | PSNR(dB)↑/SSIM↑ | Params.(M)↓ | Run.(ms)↓ | MACs(G)↓ |
|---|---|---|---|---|
| EGSVR | 25.88 / 0.80 | 2.68 | 70 | 57.1 |
| RTSR | 25.59 / 0.75 | 0.06 | 4 | 1.07 |
| TS-Mamba (ours) | 27.17 / 0.82 | 3.00 | 29 | 112 |

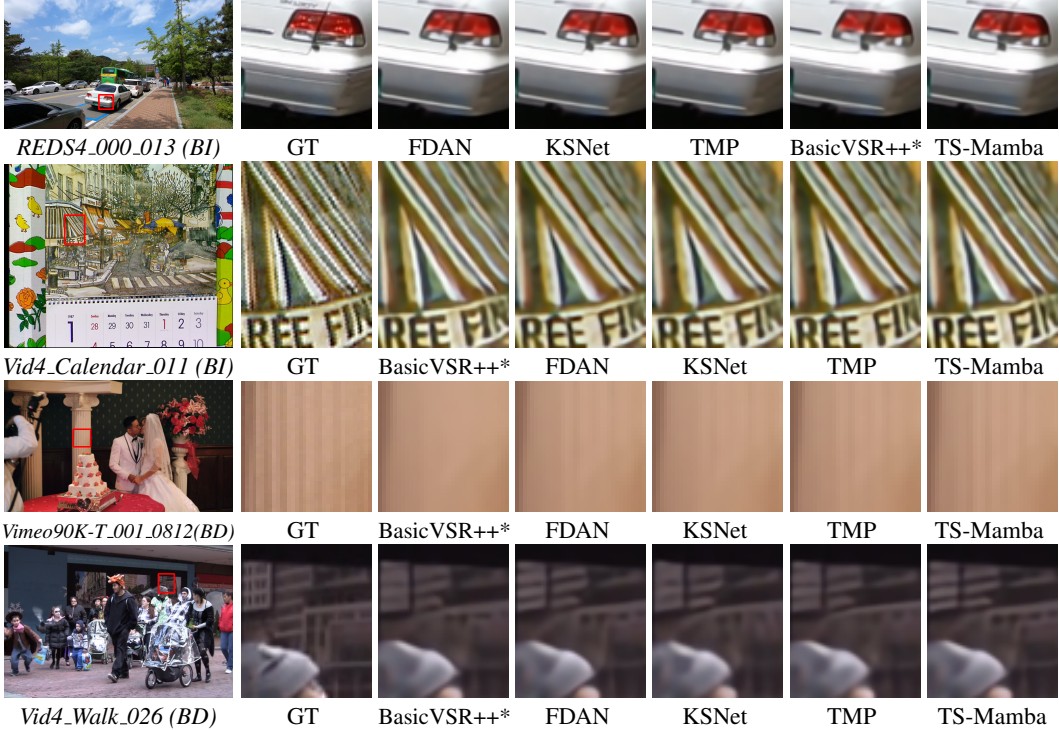

Figure 12: Visual comparison results on BI degradation (REDS4, Vid4) and BD degradation (Vimeo-90K-T, Vid4).

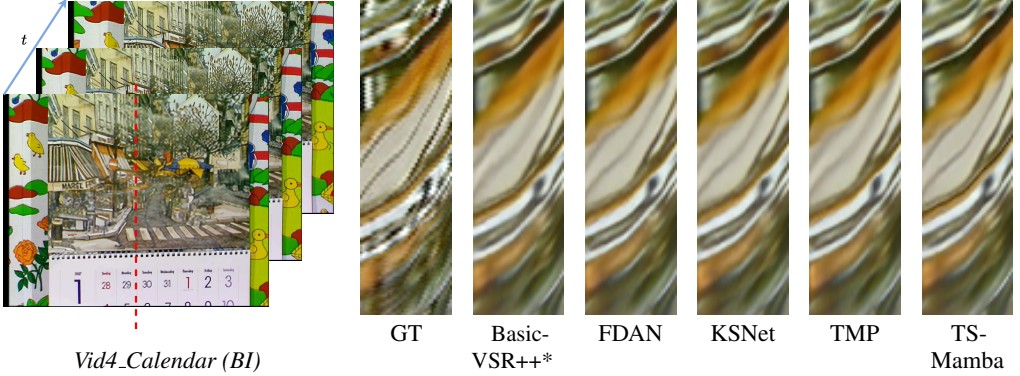

Figure 13: Temporal consistency comparison results on BI degradation for $Calendar$ video in Vid4 dataset.

performance. We provide the results of these two real-time methods with our TS-Mamba model on Vid4 dataset with BI degradation in Table 11. Noted that RTSR achieves the lowest complexity and fastest inference speed but has the unsatisfactory SR performance.

*(5) Additional Visual Results*. To provide more comprehensive comparison, we compare our TS-Mamba with four state-of-the-art online VSR methods, i.e., BasicVSR++*, FDAN (Yang et al., 2023), KSNet (Jin et al., 2023), and TMP (Zhang et al., 2024b) in visual results and temporal consistency of restored high-resolution videos.

We first provide more visual results on three test datasets, i.e., REDS (Nah et al., 2019), Vid4, and Vimeo-90K-T (Xue et al., 2019) on BI and BD degradations. These results are illustrated in Figure 12. It can be found from Figure 12 that our TS-Mamba model achieves better visual results than other online VSR methods.

Moreover, we further provide a comparison of temporal consistency. The temporal profiles of five online VSR methods and our TS-Mamba model on Vid4 $calendar$ video on BI degradation are illustrated in Figure 13. It can be found in Figure 13 that our TS-Mamba model can also achieve better temporal consistency in restored videos.

