# OpenReview forum: "Trajectory-aware Shifted State Space Models for Online Video Super-Resolution"
_ICLR.cc/2026/Conference — ICLR 2026 Poster_

### Official Review · Reviewer_7jKz · 2025-10-27

**Soundness:** 3
**Presentation:** 3
**Contribution:** 3
**Rating:** 6
**Confidence:** 4

**Summary:**

This paper proposes a trajectory-aware shifted state space model for online video super-resolution, named as TS-Mamba. TS-Mamba leverages trajectory to select relevant tokens from previous frames, shifted Hilbert scanning to compensate the spatial discontinuities, and a trajectory-aware loss to supervise trajectory generation. Experimental results show the competitive or superior of TS-Mamba compared with SOTA, with reduced complexity.

**Strengths:**

The integration of temporal trajectory into Mamba for online VSR is original.
The trajectory mechanism for video is a well-motivated and reasonable way to capture the longer temporal dependencies while maintaining low complexity. That is suitable for online video-related tasks.
Novel use of state space models (SSMs) with elaborately designed shifted operations to compensate for Hilbert scanning discontinuities，and sufficient illustrations enhance understanding and readability.
This paper is a well-organized and well-presented, with ablations and detailed training settings. The illustration of the SSMs scan is impressive and clear.

**Weaknesses:**

Author proposes a trajectory-aware shifted Mamba aggregation module to achieve the long-term spatio-temporal information aggregation. However, the existing trajectory/temporal alignment methods (e.g., deformable alignment, flow-guided deformable alignment) also can achieve this goal, a more detailed comparison with these methods would strengthen this part.

The experiments only use standard window sizes, without specific analysis on long-term dependency (e.g., varying temporal window T). A comparison between using the fixed temporal window (i.e., T=15) and other temporal window sizes would help determine the optimal size of temporal window of trajectory modeling for the proposed method.

The author did not thoroughly analyze the shortcomings of existing methods, and the description is also quite vague. It is suggested that the author conduct a more in-depth analysis of the shortcomings of existing methods to further highlight the advantages of the method proposed in this paper.

**Questions:**

(1) It is suggested to add a new set of experiments ablate the temporal window size. This is a low-effort, high-impact experiment that directly validates a core contribution of the paper.
(2) The authors should revise the introduction and related work sections to include a concise yet specific "Limitations of Prior Work" paragraph that directly motivates the design choices of their own model.

---

> ### Author Response · Authors · 2025-11-26
> **Rebuttal by Authors**
>
> **Response to Reviewer 7jKz (denoted as R4)**
> **Q4-1**: More detailed comparison with existing trajectory/temporal alignment methods (e.g., deformable alignment, flow-guided deformable alignment).
> **A4-1**: Thank you for the suggestions. We provide the detailed comparisons in model structures and experimental results.
> (1) Structures. Current online VSR methods mainly rely on CNN-based modules, such as deformable attention pyramid in DAP [WACV2023], flow-guided deformable attention in FDAN [ICIP 2023], and flow-guided deformable alignment in BasicVSR++ [CVPR2022]. These methods are constrained by **short-term modeling** and **local receptive fields**, limiting their performance potential. TS-Mamba overcomes these limitations by utilizing video trajectories and efficient shifted SSM blocks with **global receptive fields** to enable **long-range aggregation**, thereby advancing online VSR capabilities.
> (2) Results. We implement four representative alignments (defroamable convolution network (DCN), flow-guided deformable alignment (FGDA), deformable attention (DA) and flow-guided deformable attention (FDA)) to replace our TSMA module in TS-Mamba for a **comprehensive comparison**. The results are reported (see table). TSMA module achieves the **best performance** and the **lowest complexity**, which demonstrates its advantages.
>
> Models|PSNR/SSIM|Params.(M)|Run.(ms)|MACs(G)
> | :--- | ---: | :---: |  :---: | --- |
> DCN|30.59/0.8696|**2.8**|30|132
> FGDA|30.64/0.8701|3.1|42|148
> DA|30.67/0.8706|3.0|33|145
> FDA|30.69/0.8714|3.0|31|142
> TSMA (ours)|**30.73/0.8727**|3.0|**29**|**112**
>
> **Q4-2**: Comparison on the different temporal window sizes.
> **A4-2**: Thank you for the comments. We ablate the temporal window size $T$ from 3 to 23 and find that performance improves but saturates at $T$=19, beyond which **complexity rises with minimal gain**. Some previous works, e.g., FDAN,TMP, use the $T$=15 as the fixed window size, we also set our temporal window size $T$ as 15 in our experiment when we design our model for **fair comparison**. Therefore, we set the temporal window size as 15 in our TS-Mamba.
>
> Models|PSNR/SSIM|Params.(M)|Run.(ms)|MACs(G)
> | :--- | ---: | :---: |  :---: | --- |
> $T$=3|30.57/0.8698|2.4|23|96
> $T$=7|30.65/0.8713 |2.7|25|103
> $T$=11|30.70/0.8722 |2.8|27|107
> $T$=15 (ours)|30.73/0.8727|3.0|29|112
> $T$=19|30.73/0.8730 | 3.3|31|118
> $T$=23 |30.74/0.8731| 3.7|33|123
>
> **Q4-3**: It is suggested that the author conduct a more in-depth analysis of the shortcomings of existing methods to further highlight the advantages of the method proposed in this paper.
> **A4-3**: Thank you for the comments. We discussed the shortcomings of existing methods, including deformable-based, Mamba-based, and diffusion-based methods in our revision (**Section 1, 2.1, 2.2, 2.3**).
> (1) Deformable-based. DAP, and FDAN design the deformable-based modules to model the temporal alignment/aggregation. **Only use one previous frame** limits the ability of deformable-based modules while our TS-Mamba introduces the trajectories to select token from **all previous frames**, enabling our TSMA module to **utilize more effective information** for online VSR.
> (2) Mamba-based methods. While methods such as VSRM, MamEVSR compromise efficiency with **complex scanning strategies**, our TS-Mamba maintains **local spatial continuity** in Mamba through shift operations under Hilbert scannings, improving VSR performance **without compromising model efficiency**.
> (3) Diffusion-based methods. Unlike these diffusion-based methods (LiftVSR, UltraVSR, FlashVSR) that densely aggregate all long-range information at **high cost**, TS-Mamba employs trajectory-based token selection in shifted SSMs to **maintain efficiency** while achieving **high-performance** long-range modeling for online VSR.
>
> **Q4-4**: It is suggested to add a new set of experiments ablate the temporal window size. This is a low-effort, high-impact experiment that directly validates a core contribution of the paper.
> **A4-4**: Thank you for your suggestions! We have conducted experiments of the temporal window size and provided the results in the table of answer **A4-2**. As performance gains diminish beyond $T$=19 while complexity rises, and for fair comparison with methods like FDAN, TMP, for a fair comparison, we adopt $T$=15 as the temporal window size for TS-Mamba.
>
> **Q4-5**: Revise the introduction and related work sections to include a ``Limitations of Prior Work" paragraph that directly motivates the design choices of their own model.
> **A4-5**: Thanks for the suggestions! We rewrote the second and third paragraphs of the **introduction Section** to **emphasize the limitations** of the previous works and to point out **our motivation** in the **third paragraph** of our revised paper. Moreover, we also outlined the limitations of prior works in related work Section (**Section 2.2**) to introduce our motivation. The complete descriptions are presented in our revised paper.

---

### Official Review · Reviewer_zoqq · 2025-10-31

**Soundness:** 2
**Presentation:** 2
**Contribution:** 2
**Rating:** 4
**Confidence:** 4

**Summary:**

This paper focuses on online video super-resolution (VSR) and proposes a new model called TS-Mamba, which is built upon State Space Models (SSMs). The method introduces long-term trajectory modeling combined with the lightweight Mamba framework to improve spatio-temporal information aggregation efficiency in video sequences.

**Strengths:**

The integration of Mamba-based spatial modeling with trajectory-aware design is conceptually clear and implemented in a well-structured manner.

The proposed Trajectory-aware Shifted Mamba Aggregation (TSMA) module is technically sound and engineering-wise elegant.

The model achieves reasonable complexity-efficiency trade-offs — it incorporates long-term temporal modeling while maintaining relatively low MACs and parameter counts compared to prior online VSR baselines.

**Weaknesses:**

1. Biased motivation.

The paper claims that “most existing VSR methods solely employ one neighboring previous frame”, yet this statement overlooks numerous recent works that already explore long-range temporal modeling. For instance:

[1] Wang, Xijun, et al. LiftVSR: Lifting Image Diffusion to Video Super-Resolution via Hybrid Temporal Modeling with Only 4×RTX 4090s. arXiv:2506.08529 (2025).

[2] Liu, Yong, et al. UltraVSR: Achieving Ultra-Realistic Video Super-Resolution with Efficient One-Step Diffusion Space. arXiv:2505.19958 (2025).

[3] Zhuang, Junhao, et al. FlashVSR: Towards Real-Time Diffusion-Based Streaming Video Super-Resolution. arXiv:2510.12747 (2025).
These studies demonstrate substantial progress in long-term temporal modeling, undermining the paper’s claim that such research is scarce. This weakens the motivation and positioning of the work.

2. Lack of analysis on real-time VSR methods.

In lines 109–117, the authors argue that optical-flow or deformable-convolution-based methods are computationally expensive. However, multiple studies have achieved real-time performance with efficient architectures, such as:

[1] Cao, Yanpeng, et al. Real-Time Super-Resolution System of 4K-Video Based on Deep Learning. IEEE ASAP 2021.

[2] Jiang, Y., Nawała, J., Feng, C., et al. RTSR: A Real-Time Super-Resolution Model for AV1 Compressed Content. IEEE ISCAS 2025.
The paper fails to include any quantitative or empirical comparison with these real-time approaches and instead relies on generic qualitative claims. This lack of evidence weakens its argument regarding computational superiority.

3. Outdated comparison on standard benchmarks.

The experiments are conducted on REDS4, Vid4, and Vimeo-90K-T datasets, but do not include recent state-of-the-art models that have achieved higher PSNR and SSIM. For example:

Lin, Jing, et al. Unsupervised Flow-Aligned Sequence-to-Sequence Learning for Video Restoration. ICML 2022 — achieving 31.96 dB and 37.63 PSNR on REDS4 and Vimeo-90K-T, outperforming the proposed method.

Dong, Shuting, et al. DFVSR: Directional Frequency Video Super-Resolution via Asymmetric and Enhancement Alignment Network. IJCAI 2023.
Without comparisons to these recent methods, the experimental evaluation is incomplete and fails to convincingly establish the method’s competitiveness.

Limited novelty in applying Mamba to VSR.
The use of Mamba-based State Space Models for video super-resolution is not new.

Event-based Video Super-Resolution via State Space Models (CVPR 2025) already introduced a similar Mamba framework for VSR.
The authors do not clarify how TS-Mamba fundamentally differs from or advances beyond these prior Mamba-based designs. Thus, the methodological contribution appears incremental rather than novel.

**Questions:**

N/A

---

> ### Author Response · Authors · 2025-11-26
> **Rebuttal by Authors**
>
> **Response to Reviewer zoqq (denoted as R3)**
> **Q3-1**: Biased motivation of long-range temporal modeling.
> **A3-1**: Thank you for the comments.  We provide comparisons and clarify our novelty below.
> (1) Online VSR. TS-Mamba addresses the online VSR task, where methods must utilize **only previous frames** under strict latency and complexity constraints. Current online VSR methods are notably restricted to **one previous frame** due to complexity limitation.
> (2) Comparison of long-term modeling. Three mentioned works (LiftVSR (no released code), UltraVSR, FlashVSR) use the all long-range information in videos, leading to **high complexity and temporal redundancy**. TS-Mamba introduces trajectory and designs efficient shifted SSMs to select the most similar token for efficient long-range spatio-temporal aggregation.
> (3) Novelty. TS-Mamba is **the first SSMs-based online VSR model** and **the first time** to introduce video trajectories into Mamba, which establishes its uniqueness. The above analysis and discussions confirm both the necessity and advantages of these designs.
> (4) Results. We provide results on REDS4 dataset in following table. TS-Mamba achieves the **highest** PSNR and SSIM by a large margin, alongside **significant model efficiency**, demonstrating its significant advantages.
>
> Methods|PSNR$\uparrow$|SSIM$\uparrow$|Param.(M)$\downarrow$|Run.(ms)$\downarrow$
> | :--- | ---: | :---: |  :---: | --- |
> LiftVSR|24.34|-|-|-
> UltraVSR|24.50|0.6962|10.5|89
> FlashVSR|24.11|0.6511|1780.14|15500
> TS-Mamba (ours)|**30.73**|**0.8727**|**3.1**|**29**
>
> **Q3-2**: Lack of analysis on real-time VSR methods.
> **A3-2**: Thank you for the suggestions. We provide analysis of deformable-based methods and real-time VSR methods below.
> (1) Analysis of deformable. We implement **four representative** alignments, i.e., deformable convolution network (DCN), flow-guided deformable alignment (FGDA), deformable attention (DA), and flow-guided deformable attention (FDA), to replace the TSMA module in TS-Mamba (see table). Results demonstrate the **advantages** of TSMA module in performance and efficiency.
> Models|PSNR/SSIM|Params.(M)|Run.(ms)|MACs(G)
> | :--- | ---: | :---: |  :---: | --- |
> DCN|30.59/0.8696|**2.8**|30|132
> FGDA|30.64/0.8701|3.1|42|148
> DA|30.67/0.8706|3.0|33|145
> FDA|30.69/0.8714|3.0|31|142
> TSMA (ours)|**30.73/0.8727**|3.0|**29**|**112**
>
> (2) Analysis of real-time methods. Two real-time methods (EGVSR,RTSR) only use CNN networks with **local receptive field** and neglect the long-rang temporal information while TS-Mamba introduces the **low-complexity and global receptive field** Mamba with temporal trajectories to achieve the high quality long-range spatio-temporal aggregation.
> (3) Results. We provide the results on Vid4 dataset in following table.  Noted that RTSR achieves the lowest complexity and fastest speed but has the **unsatisfactory SR performance** while TS-Mamba has the advantages of VSR performance and real-time efficiency.  We also provide the discussions and comparisons in our supplementary (Section 7.2.2 (4))
> Methods|PSNR/SSIM|Params.(M)|Run.(ms)|MACs(G)
> | :--- | ---: | :---: |  :---: | --- |
> EGSVR|25.88/0.80|2.68|70|57.1
> RTSR|25.59/0.75|0.06|4|1.07
> TS-Mamba (ours)|**27.17/0.82**|3.00|29|112
>
> **Q3-3**: Outdated comparison on standard benchmarks.
> **A3-3**:  Thank you for the suggestions! We provide the comparisons below.
> (1) Two VSR methods (S2SVR, DFVSR (no released code)) use the consecutive one or two frames to design the **high-complexity modules** for temporal alignment, while TS-Mamba achieves an efficient trajectory-aware long-range spatio-temporal aggregation, enabling it applicable in online VSR.
> (2) Following the suggestions, we conduct the experiments of two mentioned VSR methods (S2SVR, DFVSR), and two latest generic VSR methods (MIA-VSR [CVPR2024], IA-RT [CVPR2024]). Due to the space limitation, the partial results of four VSR methods are provided in following table, complete table is provided in **Table 1** of our revision.
>
> Methods|PSNR/SSIM|Params.(M)|Run.(ms)|MACs(G)
> | :--- | ---: | :---: |  :---: | --- |
> S2SVR|31.96/0.8988|13.4|194|3462
> DFVSR|32.76/0.9081| 7.1|-|-
> MIA-VSR| 32.78/0.9220|16.5|180|3220
> IA-RT|32.90/0.9138|13.4|223|5020
> TS-Mamba (ours)|30.73/0.8727|**3.0**|**29**|**112**
>
> **Q3-4**:  Limited novelty in applying Mamba to VSR.
> **A3-4**:  Thanks for the suggestions! Two Mamba-based methods (MamEVSR, VSRM) **repeatedly use multiply scannings** to enhance ability of Mamba, leading to high complexity and limited performance. TS-Mamba introduce shift operations for Hilbert scannings to enhance the ability of Mamba to maintain local spatial continuity, achieving high performance and efficiency.  As the **first Mamba-based online VSR model** and a pioneering effort in integrating video trajectories into Mamba, TS-Mamba demonstrates **unique design advantages**. We also emphasize the our novelty in our revised paper (Section 1 and Section 2.2, 2.3).

---

> ### Author Response · Authors · 2025-11-27
> **Further discussion with Reviewer zoqq**
>
> Dear Review zoqq,
>
> We sincerely appreciate the time and effort you devoted to reviewing our work, as well as your thoughtful and insightful comments. We have carefully considered your feedback and provided detailed responses to your concerns:
>
> 1. We clarify our novelty and provide more comparisons for long-range temporal modeling.
> 2. We analysis and compare the recent real-time methods.
> 3. We provide results and comparisons of recent VSR methods in our revision.
> 4. We compare the recent Mamba-based methods and further clarify our novelty to strengthen the uniqueness of our designs.
>
> As the author–reviewer discussion phase is nearing its conclusion, we would like to confirm whether our responses have addressed your concerns. If you have any questions or suggestions, please do not hesitate to let us know.
>
> Thank you again for your thoughtful feedback and kind support.
>
> Best regards,
>
> Authors

---

### Official Review · Reviewer_BFtp · 2025-10-31

**Soundness:** 3
**Presentation:** 3
**Contribution:** 3
**Rating:** 8
**Confidence:** 5

**Summary:**

This paper focuses on online video super-resolution which aims to reconstruct a high-resolution video from its low-resolution counterpart using only past frames, i.e., without access to information from future frames. This research has high industrial values, especially for real-time and streaming applications. In this paper, authors propose a variant of Mamba models, called TS-Mamba, to model the long-term temporal information bewtween frames. Specifically, to effectively leverage similar features from previous frames, the proposed methods aims to select similar features along the trajectory. This token selection strategy keeps spatial consistency and avoids artifacts in reconstruction. The original Mamba models efficiently process 1D sequential data, but the Raster scan mechanism cannot handle image data, especially in boundary areas. To this end, TS-Mamba introduces a variant of Hilbert scan in the spatial domain, which is restricted within and between local windows. By incoorporating the shifting operator, the proposed model can effectively process image content from adjacent windows for restoration. To better supervise model during training, it additionally propsoes trajectory loss. The experiment results have demonstrated promising results in REDS4 and Vimeo-90K datasets with BI degradation and BD degradation. Overall, this is a good paper. Unlike previous methods, it introduce Mamba structure for online video super-resolution and has demonstrated a better trade-off between restoration quality and speed. However, I still have several suggestions, questions and confusions about this paper listed as below.

**Strengths:**

1. The paper comprehensively discusses the related works in online video super-resolution and points out the existing limitations of the existing methods.
2. TS-Mamba solely utilizes past frames for online video super-resolution and aggregates long-range information through trajectory-aware token selection, which motion paths across multiple previous frames.
3. The proposed method is efficient without sacrificing quality, leading to a better trade-off between reconstruction quality and processing time.

**Weaknesses:**

1. what is $$v_{\tau_{i}^{h_{j}}}? Are they math typos in Eq.(7) and Eq.(8)?
2. The method proposes a trajectory-aware method and define a temporal trajectory among video frames in Eq.(3). However, I am confused on this definition. It is better to elaborate more on what positions among video frames belong to the same trajectory.
3. The method introduces a dual-path block, i.e., intra-window compensation branch and inter-window compensation branch. What is the key difference between two blocks? Why one block can handle intra-window content and another one can hanld inter-window content?
3. The method proposes variant scanning strategies. What scanning strategy it adopts in experiment? And why using this scanning strategy instead of other strategies?
4. It is better elaborate on the descriptions of shifted SSMs block, especially for math notations. It seems that elimintation value is a hyper-parameter. Does it matter for the restoration performance?
5. In the paper, it proposes trajectory-aware loss. How to compute this loss? Does it compute L2 loss between the trajectories in LR and the corresponding counter in HR images?
6. It is better to demonstrate visual results associated with Table 2. However, due to page limited, it is still acceptable.

**Questions:**

I include the questions and concerns on this paper in Weakness section. Please authors prepare their rebuttal reference to the questions listed in Section 6.

---

> ### Author Response · Authors · 2025-11-26
> **Rebuttal by Authors**
>
> **Response to Reviewer BFtp (denoted as R2)**
> **Q2-1**: $v_{\tau_i^{h_j}}$ in Eq.(7) and Eq.(8).
> **A2-1**: Thank you for the comments. $v_{\tau_i^{h_j}}$ is the selected token from previous frames, where $\tau_{i}$ represents $i^{th}$ trajectory and $h_j\in[1,T-1]$ represents the indices of the selected token for previous frames. In this work, we use $h_j$ to represent **the indices of the selected tokens** for distinguish with commonly tokens indices $k$ in Eq.(6). Therefore, $v_{\tau_i^{h_j}}$ represents the selected token from previous frames.
> **Q2-2**: Definition of temporal trajectory and positions among video frames belong to same trajectory.
> **A2-2**: Thank you for the comments. We clarify definition of trajectory and point out the positions of same trajectory below.
> (1) Definition of trajectory. Token $\\{q_i^t\\}$ determinate the representation of trajectories in Eq.(3) and trajectories are updated based on coordinate map of trajectories and optical flow between frames (as mentioned in Section 4.1).
> (2) Positions. Coordinate map of trajectories is formulated as $\mathcal{M}^t$=[($x_i^t$, $y_i^t$)]$\_{H\times W}$, $i\in[1,N]$, where $H$ and $W$ are height and width of feature. Updated coordinate map is denoted as:  $^{*}\mathcal{M}^t$=$\mathrm{S}(\mathcal{M}^t, f^{t+1})$, where $\mathrm{S}(\cdot)$ is spatial sampling, $f^{t+1}$ is optical flow between time $t$ and time $t+1$. Therefore, positions among video frames belongs to same trajectory can be found.
> **Q2-3**: Key difference between IntraWCB and InterWCB and respective reasons.
> **A2-3**: Thank you for the comments. We clarify them below.
> (1) Key difference. IntraWCB and InterWCB are distinguished only by **their implemented shift operations**. The respective shift operations dictate the function of each branch. For example, under first scanning is Scan-1, IntraWCB and InterWCB implement shift operations $U(1)$ and $UL(3)$, both branches perform under the same scanning Scan-3.
> (2) Reasons. Two parallel S-SSMs blocks are designed to **independently compensate** for intra-window loss and inter-window loss, whereby **their merging** achieves the **interactive compensation**. This structure not only enables the model to focus on specific losses but also reduces inference time than serial design.
> **Q2-4**:Adopted scanning strategy and reasons.
> **A2-4**:Thanks for the comments. (1) Adopted scannings. We design specific procedures for intra-window and inter-window discontinuity eliminations. When the scanning of standard SSMs block is determined, shift operations and scannings in IntraWBC and InterWBC are determined **based on the best elimination**.
> (2) Reasons. In Section 3.4 and Section 7.1, we list representative procedures and analysis their elimination values to find out the best procedures for intra-window and inter-window discontinuities. Based on this, we determinate two procedures to construct TSMA module for sufficient elimination of discontinuity. Scanning strategy is determined based on elimination values of procedures, with objective of achieving **optimal elimination of discontinuities**.
> **Q2-5**: Elaborate on descriptions of shifted SSMs block. Does elimination value matter for the performance?
> **A2-5**: Thank you for your suggestions! We formulate the procedure as defined in Eq. (9) of our revised paper:
> $\mathcal{P}(l,\mathcal{S}f(p),j)$=$\mathcal{S}c_1(l)\rightarrow\mathcal{S}f(p) \rightarrow \mathcal{S}c_2(j)$ and calculate the value range of elimination value $\delta$, $\delta_\mathrm{intra}$ and $\delta_\mathrm{inter}$. Based on their value range, we **find out the procedures** of best $\delta_\mathrm{intra}$ and best $\delta_\mathrm{inter}$ to construct our shifted SSMs. Since elimination value is determined by procedure, it further indicates the restoration performance.
> **Q2-6**: How to compute this loss? Does it compute L2 loss between the trajectories in LR and counter in HR images?
> **A2-6**: Thank you for the comments. Trajectories of LR video and HR video can be generated by Eq.(3). HR trajectories are formulated as: $\mathcal{T}^t_{HR}$=$\\{\tau_{i(HR)}^{k}=(x_i^k,y_i^k)\\}$, $i \in[1,M], k\in[t-T,t]$. In $\mathcal{T}^t$ and $\mathcal{T}^t_{HR}$, each trajectory contains a sequence of coordinates. HR trajectories are downsampled and reduce elements of trajectories to ensure consistency in resolution and value range with LR trajectories, obtaining the **downsampled trajectories** $((\mathcal{T}^t_{HR})\downarrow_{\hat{s}})/\hat{s}$. $L_2$ loss measures the distance between two LR trajectories which quantifies the positional difference to supervise the trajectory generation.
> **Q2-7**: Visual results associated with Table 2.
> **A2-7**: Thank you for the suggestions! We added visual results associated with Table 2 in revised paper (in **Figure 5** and **Section 4.3**). The visual results presents the contributions of our designs, which further demonstrates their effectiveness.

---

### Official Review · Reviewer_2xrU · 2025-10-31

**Soundness:** 3
**Presentation:** 3
**Contribution:** 3
**Rating:** 4
**Confidence:** 4

**Summary:**

This paper proposes TS-Mamba, a state-of-the-art method for online video super-resolution (VSR), leveraging trajectory-aware shifted state space models (SSMs) for efficient spatio-temporal information aggregation. The model addresses challenges in long-term temporal modeling for real-time applications while keeping computational complexity low. The approach combines token-level spatio-temporal aggregation with a novel trajectory-aware shifted Mamba aggregation (TSMA) module. TS-Mamba constructs trajectories within video frames to select similar tokens from previous frames and aggregates them using shifted SSM blocks. This enables improved video frame restoration with reduced computational overhead. The proposed model is evaluated on several benchmark datasets (REDS, Vimeo-90K-T, Vid4) and outperforms five state-of-the-art online VSR methods in terms of PSNR/SSIM while reducing complexity by over 22.7%.

**Strengths:**

1. Performance: The model achieves superior performance in terms of PSNR/SSIM and visual quality across multiple benchmark datasets (REDS, Vid4, Vimeo-90K-T) and degradation types (BI and BD), demonstrating its robustness in real-world video restoration scenarios.

2. Computational Efficiency: TS-Mamba successfully reduces complexity by 22.7% in terms of MACs compared to existing methods, making it a strong candidate for real-time online VSR applications. The model is also one of the fastest among the tested methods.

3. Comprehensive Ablation Study: The authors provide an extensive ablation study validating the importance of trajectory-aware components and the shifted SSM blocks, as well as the impact of different design choices (e.g., token number, shift operations). This strengthens the validity of their claims and helps illustrate the contributions of each part of the model.

**Weaknesses:**

1. Lack of Comparison. The proposed scheme fails to discuss or compare with several recent restoration schemes that leverage state-space models (SSMs, e.g. Mamba) for superior performance. For instance, MambaIR and MambaIRv2 introduced a residual Mamba-based backbone (with convolution and channel attention) to capture global dependencies in image super-resolution and denoising, outperforming a SwinIR Transformer baseline. The more recent TAMambaIR improves efficiency by modulating the state-space transition for complex textures and using multi-directional scanning, achieving state-of-the-art results across image restoration tasks (e.g. super-resolution, deraining, low-light enhancement). In the video domain, VSRM proposes dual Spatial-to-Temporal and Temporal-to-Spatial Mamba blocks for long-range spatio-temporal feature extraction and a deformable cross-Mamba alignment module for flexible frame alignment, yielding new state-of-the-art performance on VSR benchmarks. Likewise, The authors should incorporate and evaluate these methods to ensure a comprehensive comparison with the current state-of-the-art in video super-resolution. （I believe that the VSR method, when transformed into the online VSR setting, can be compared.）

2. Insufficient Theoretical Justification: While the paper presents an innovative approach, the theoretical justification for the trajectory-aware shifted state space model is somewhat lacking. Specifically, a formal analysis of the long-term spatio-temporal aggregation process and a comparison with existing models using similar principles (e.g., in flow-guided deformable attention models) would provide more clarity. Additionally, the authors mention the use of shifted operations to enhance spatial continuity but could further explore the theoretical implications of this operation on model behavior.

3. Insufficient Validation. A notable weakness is the absence of experiments on real-world VSR datasets. The method’s results are only reported on standard synthetic benchmarks (REDS4, Vid4, Vimeo-90K) with bicubic or simulated blur degradation, but no evaluations on real degraded videos were provided . As real applications involve unknown and complex degradations (compression artifacts, sensor noise, motion blur, etc.), the paper should have validated the approach on established real-world video SR benchmarks. For example, testing on datasets like RealVSR (ICCV 2021), which is built with a dual-camera system on the iPhone 11 Pro Max, would demonstrate the model’s robustness to in-the-wild conditions

4. Potential on Certain Datasets: The model performs exceptionally well on the benchmarks but lacks a more nuanced discussion of failure cases or scenarios where the method might struggle. A brief mention of potential limitations (e.g., when frames are highly dynamic or occlusions are prevalent) would enhance the robustness of the paper's claims.

**Questions:**

See the weakness.

---

> ### Author Response · Authors · 2025-11-26
> **Rebuttal by Authors**
>
> **Response to Reviewer 2xrU (denoted as R1)**
> **Q1-1**: Comparisons of Mamba-based methods.
> **A1-1**: Thanks for the suggestions. We provide the comparisons in structures and results below.
> (1) Structures. Mentioned Mamba-based methods (MambaIR, MambaIRv2, TAMambaIR, VSRM) use multiply scannings or an extra interaction module to enhance the ability of Mamba, but they neglect the local spatial continuity of Mamba, which results in the limited SR performance. Our TS-Mamba analyzes the local spatial discontinuity, and combines Hilbert scannings with shift operations to strengthen ability of Mamba.  **Shift operations is a simple yet efficient way** to strengthen local spatial continuity of Mamba without increasing complexity, which effectively helps TS-Mamba achieve high efficiency in spatio-temporal aggregation.
> (2) Results. We implement **a transformed method** (VSRM*) for online VSR and results are provided in following table. Although VSRM has better performance than other methods, its high complexity makes it unsuitable for online applications. Compared with VRSM*, our TS-Mamba achieves a high PSNR and low complexity due to long-range spatio-temporal aggregation and shifted Mamba compensation. VRSM* only uses two previous frames to aggregate temporal information, limiting the exploration of long-range spatio-temporal information.  **Complete results** are provided in **Table 1** of the revised paper.
>
> Methods|PSNR/SSIM|Params.(M)|MACs(G)
> | :--- | ---: | :---: | --- |
> MambaIR|32.25/0.9019|20.42|779.7
> MambaIRv2|32.48/0.9054|23.10|1567.2
> VSRM|33.11/0.9162|17.1|2174
> VSRM*|30.64/0.8701|3.1|136
> TS-Mamba (ours)|30.73/0.8727|**3.0**|**112**
>
> **Q1-2**: Insufficient Theoretical Justification.
> **A1-2**: Thanks for the comments.  We clarify the theoretical justification below.
> (1) Analysis of the long-term spatio-temporal aggregation. The long-term spatio-temporal aggregation in TSMA module is built on the trajectory theory of video [TTVSR,CVPR2022]. Trajectories is constructed based on the generated tokens and is updated by motion between frames. Based on the trajectories, TSMA module **selects the most similar tokens** from previous frames and then adopts the shifted Mamba block to aggregate the long-term spatio-temporal tokens. We introduce **the specific shift operations based on Hilbert scannings** to achieve the compensation for local spatial continuity of Mamba in a simple yet efficient manner.
> (2) Comparison with similar principles. FDAN [ICIP2023] design a flow-guided deformable attention module that applied in a short-term spatio-temporal aggregation **(only two frames)**, which limits the long-term modeling. In our TS-Mamba, the long-term spatio-temporal information is explored based on trajectories and then is aggregated by shifted Mamba block, which improves the VSR performance. Moreover, our TSMA module is performed to search most similar tokens for different regions will be aggregated, which mitigates the restriction of the whole frame process and **avoids the computational costs** in optical flow estimation.
> (3) Shift operations. Mamba converts 2D images into 1D tokens for liner complexity, resulting in spatial discontinuous inherent to images. To eliminate the discontinuous, we **calculate the elimination value** with and without shift operations under Hilbert scannings to guide the compensation (details in our supplementary (Section 7.1 and Figure 10)).
> **Q1-3**: Insufficient Validation on real-world VSR datasets.
> **A1-3**: Following reviewer's suggestions, we conduct experiments for VSRM, VSRM*, TS-Mamba and three real-world VSR methods, i.e., RealVSR [ICCV2021], BasicRealVSR [CVPR2022], RealViformer [ECCV2024] on RealVSR dataset to evaluate on **real-world scenarios**. The corresponding results are provided in the following table.
> It is found that the general VSR results are not as effective as the RealVSR methods, since no noise or unknown complex degradations were introduced during the training process. Compared to RealVSR, TS-Mamba achieves better VSR results while maintaining lower complexity, which indicts that TS-Mamba has robustness for in-the-wild scenarios and makes it **a potential replacement for RealVSR**.
> Methods|ILNIQE↓/NRQM↑|Params.(M)|Run.(ms)
> | :--- | ---: | :---: | --- |
> RealVSR|34.39/3.795|**2.7**|772
> BasicRealVSR|30.37/6.582|6.3|73
> RealViformer|28.61/6.588|5.3|49
> VSRM|30.29/6.613|17.1|223
> VSRM*|33.29/4.368|3.1|31
> TS-Mamba (ours)|32.54/5.161|3.0|**29**
>
> **Q1-4**: Potential on Certain Datasets.
> **A1-4**: Thanks for your comments. We provide a failure case when highly dynamic rotation occurs in **Figure 6** and added **Limitations Section** in our **revision**. The generated trajectories are inaccurate when **highly dynamic rotation occurs** and rotation information cannot be reconstructed, thus limiting the VSR performance. Due to the high difficulty of modeling rotation, other online VSR methods also fail to obtain complete rotation information.

---

> > ### Author Response · Authors · 2025-11-27
> > **Further discussion with Reviewer 2xrU**
> >
> > Dear Review 2xrU,
> >
> > Thank you for taking the time to review our paper and for your thoughtful and constructive comments. We have carefully considered your comments and provided detailed responses to your questions:
> >
> > 1. We provide thorough comparisons with Mamba-based methods in model structures and experimental results.
> > 2. We clarify detailed theoretical justification for the design of our TSMA module and the shift operations within the shifted SSMs block to strengthen the theoretical foundation of our work.
> > 3. We conduct the experiments and analysis on the real-world dataset, comparing our method with three representative real-world VSR methods and two Mamba-based VSR methods.
> > 4. We provide a failure case when highly dynamic rotation occurs and add the Limitations Section in our revised paper.
> >
> > As the author–reviewer discussion phase is nearing its conclusion, we would like to confirm whether our responses have addressed your concerns. If you have any questions or suggestions, please let us know. We would be grateful to hear them.
> >
> > Thank you again for your time, effort, and valuable feedback.
> >
> > Best regards,
> >
> > Authors

---

### Author Response · Authors · 2025-11-28
**Summary of the Authors' Responses**

Dear area chairs and reviewers,

We sincerely thank area chairs and reviewers for their valuable time and thoughtful comments. This work proposes a Trajectory-aware Shifted SSMs (TS-Mamba) for online VSR, leveraging both long-term trajectory modeling and low-complexity Mamba to achieve efficient spatio-temporal information aggregation. TS-Mamba is **the first time** to introduce video trajectories into Mamba and **the first SSMs-based online VSR model**, and it establishes a superior trade-off between restoration quality and speed.

To assist the newly assigned area chairs and help reduce their workload, we summarize the strengths highlighted by the reviewers and key points of our responses below.

We are pleased to receive positive feedback:
1. **All reviewers** (R1–2xrU, R2–BFtp, R3–zoqq, R4–7jKz) recognized our superior performance and computational efficiency (**over 22.7% complexity reduction** in terms of MACs).
2. **R1** recognized the contributions from validated ablation study and **R3** recognized clear methodology and well-structured design.
3. **R2** and **R4** acknowledged our original novelty, the high research values and well paper organization.

We have provided point-by-point responses to each reviewer's comments below and have uploaded a revised paper. We offer a summary of our revision as follows:

1. We compared our approach with **more recent methods** in methodology and experimental evaluation, including Mamba-based, Diffusion-based, Transformer-based, and other CNN-based methods.
2. We provided **the theoretical justification** for the trajectory generation, the proposed TSMA module and the shifted SSMs block.
3. We clarified **the novelty** of our method and its distinctions from existing methods.
4. We performed **additional ablation studies**, including those on temporal alignment/aggregation, long-range temporal modeling, real-world dataset, real-time methods, temporal window size, and recent advanced structures.
5. We demonstrated the **advantages** of the proposed TSMA module through experiments.
6. We elucidated the **descriptions** and **mathematical notations** of the shifted SSMs block.
7. We provided **more visual results** for the ablation studies and failure cases.
8. We outlined the **limitations** of our method and incorporated the corresponding section to discuss them.

We are confident and hopeful that our responses can address the concerns of each reviewer and area chair. We are deeply grateful to the area chairs and reviewers for their dedicated effort and excellent work. Their insightful feedback has further strengthened our paper. The authors offer their sincere respect and appreciation to all involved!

Best regards,

Authors

---

### Meta-Review · Area_Chair_UtET · 2026-01-02

**Summary:**

This paper presents TS-Mamba, a novel trajectory-aware shifted state space model for online video super-resolution. The method leverages long-term trajectory modeling combined with low-complexity Mamba architecture to achieve efficient spatio-temporal information aggregation for real-time applications.

The reviewers raised several important concerns that informed the decision-making process:

Reviewer 2xrU questioned the lack of comparisons with recent Mamba-based methods, insufficient theoretical justification, and absence of real-world dataset validation.

Reviewer BFtp sought clarification on mathematical notations and trajectory definitions.

Reviewer zoqq challenged the motivation as potentially biased and noted missing comparisons with real-time methods and recent benchmarks.

Reviewer 7jKz  requested more detailed comparisons with existing alignment methods and analysis of temporal window sizes.

All reviewers acknowledged the paper's strengths in achieving superior performance with complexity reduction and recognized its potential contributions to the field.

**Reviewer Concerns:**

The authors have addressed the majority of reviewer concerns through their comprehensive rebuttal and revised manuscript. Key concerns successfully addressed include:

1) request for comparisons with recent Mamba-based methods, and the authors added extensive comparisons with MambaIR, MambaIRv2, TAMambaIR, and VSRM, demonstrating TS-Mamba's advantages.
2) concern about real-world validation, and experiments were added on the RealVSR dataset.
3) requests for mathematical clarification, and the authors provided detailed explanations of trajectory definitions and shifted SSM blocks.
4) concerns about biased motivation, and the authors refined their claims and added comparisons with recent diffusion-based methods.
5) request for temporal window analysis, and an ablation study varying T from 3 to 23 was conducted. The authors also added a limitations section discussing failure cases in highly dynamic rotation scenarios.

While most technical concerns have been adequately addressed, some minor issues regarding the depth of theoretical analysis and the positioning relative to all relevant baselines remain, though these do not diminish the paper's core contributions.

**Reviewer Scores:**

Based on the thorough responses and substantial revisions, I believe the reviewers would have adjusted their scores as follows:

2xrU  would likely increase to a 6 or 7, given their concerns about comparisons with Mamba-based methods and real-world validation were comprehensively addressed.

BFtp would maintain their strong acceptance score, as their technical clarification requests were fully satisfied.

zoqq would likely increase to a 6, considering the authors' extensive additions comparing with real-time methods and recent benchmarks, though some concerns about novelty positioning may persist.

7jKz  would likely increase to a 7, given the detailed ablation studies on temporal window sizes and enhanced comparisons with existing alignment methods.

The overall improved assessment reflects the authors' diligent efforts in addressing reviewer feedback and strengthening the paper's contributions to online video super-resolution.

---

### Decision · Program_Chairs · 2026-01-26

Accept (Poster)